# Mɪx-LN: Unleashing the Power of Deep Layers by Combining Pre-LN and Post-LN

**Pengxiang Li** [*]
Dalian University of Technology
lipengxiang@mail.dlut.edu.cn

**Lu Yin** [*]
University of Surrey
l.yin@surrey.ac.uk

**Shiwei Liu** [†]
University of Oxford
shiwei.liu@maths.ox.ac.uk

## Abstract

Large Language Models (LLMs) have achieved remarkable success, yet recent findings reveal that their deeper layers often contribute minimally and can be pruned without affecting overall performance. While some view this as an opportunity for model compression, we identify it as a training shortfall rooted in the widespread use of Pre-Layer Normalization (Pre-LN). We demonstrate that Pre-LN, commonly employed in models like GPT and LLaMA, leads to diminished gradient norms in its deeper layers, reducing their effectiveness. In contrast, Post-Layer Normalization (Post-LN) preserves larger gradient norms in deeper layers but suffers from vanishing gradients in earlier layers. To address this, we introduce Mɪx-LN, a novel normalization technique that combines the strengths of Pre-LN and Post-LN within the same model. Mɪx-LN applies Post-LN to the earlier layers and Pre-LN to the deeper layers, ensuring more uniform gradients across layers. This allows all parts of the network—both shallow and deep layers—to contribute effectively to training. Extensive experiments with various model sizes from 70M to 7B demonstrate that Mɪx-LN consistently outperforms both Pre-LN and Post-LN, promoting more balanced, healthier gradient norms throughout the network, and enhancing the overall quality of LLM pre-training. Furthermore, we demonstrate that models pre-trained with Mɪx-LN learn better compared to those using Pre-LN or Post-LN during supervised fine-tuning (SFT) and reinforcement learning from human feedback (RLHF), highlighting the critical importance of high-quality deep layers. By effectively addressing the inefficiencies of deep layers in current LLMs, Mɪx-LN unlocks their potential, enhancing model capacity without increasing model size. Our code is available at https://github.com/pixeli99/MixLN.

## 1 Introduction

Large Language Models (LLMs) have ushered in a new era of artificial intelligence by demonstrating unprecedented capabilities in understanding and generating human-like text (Brown, 2020; Achiam et al., 2023; Touvron et al., 2023; Dubey et al., 2024). Trained on vast datasets that span multiple languages and topics, LLMs are driving advancements across industries and academia, enhancing human-computer interactions, and fostering innovation in previously unimaginable ways.

Recent studies reveal a critical observation regarding the effectiveness of deeper layers in LLMs, particularly those beyond the middle layers. It has been shown that these deeper layers can often be pruned significantly (Yin et al., 2023), or even removed entirely (Gromov et al., 2024; Men et al., 2024), without notably affecting the model's overall capabilities. Moreover, Li et al. (2024) demonstrated that deeper layers contribute minimally to performance during fine-tuning, further

---

[*]Equal contribution.
[†]Corresponding author.

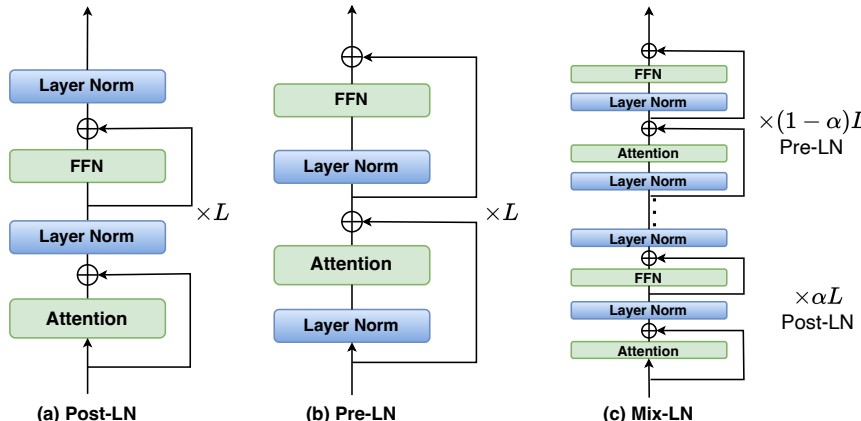

Figure 1: (a) Post-LN layer; (b) Pre-LN layer; (c) Mix-LN layer.

questioning their importance. Unfortunately, this finding has been largely overlooked by the research community, where many see it primarily as an opportunity for model compression (Siddiqui et al., 2024; Zhong et al., 2024; Sreenivas et al., 2024), rather than recognizing it as a potential shortfall in the training process.

In this paper, we seek to challenge the prevailing notion that deeper layers in LLMs are of lesser significance. The training of LLMs is extraordinarily resource-intensive, often requiring thousands of GPUs or TPUs and several months of computation on vast datasets. For example, the training of GPT-3 reportedly incurred millions of dollars in computational costs. The underutilization of deeper layers leads to inefficiencies, squandering resources that could otherwise be leveraged to enhance model performance. Ideally, all layers in a model should be well-trained, with sufficient diversity in features from layer to layer, to maximize the utility of the network's parameters (Yang et al., 2023). This makes it crucial to investigate the root causes of this underutilization and to develop strategies that fully capitalize on the potential of deeper layers, ensuring that the overall architecture is optimized for both performance and efficiency.

We hypothesize that the inefficiency of deeper layers in LLMs primarily stems from the choice of Layer Normalization. Specifically, Pre-Layer Normalization (Pre-LN) (Dai, 2019; Baevski & Auli, 2018) tends to produce smaller gradients in deeper layers, thereby diminishing their effectiveness, while Post-Layer Normalization (Post-LN) (Ba, 2016) results in larger gradients in deeper layers but leads to gradient vanishing in earlier ones. Most state-of-the-art LLMs, like GPT, LLaMA, and Mistral, employ Pre-LN, which contributes to the widespread assumption that deeper layers are inherently less effective.

To validate this conjecture, we conduct experiments with the following two categories of LLMs and compare the effectiveness of layers across different depths in Pre-LN models and Post-LN models.

- **Open-weight large-scale LLMs:** We select LLaMa2-7B (Touvron et al., 2023) as a representative Pre-LN model and BERT-large (Devlin, 2018) as a Post-LN model to evaluate the quality of their layers. Our findings confirm that the deeper layers of LLaMa2-7B exhibit high similarity, with their removal leading to minimal impact compared to the early layers. In stark contrast, BERT shows higher similarity among its first half, which contributes less to the model's output.

- **In-house small-scale LLMs:** To control for irrelevant confounding variables, we conduct a second set of experiments by training small-scale LLMs ourselves, ensuring that the only difference between the models is the choice of layer normalization. Consistent trends are observed in these experiments, reinforcing our earlier observations.

Building on these insights, we propose a novel normalization technique, dubbed `Mix-LN`, which synergizes Pre-LN and Post-LN to achieve more balanced and healthier gradient norms across the network. `Mix-LN` applies Post-LN to the earlier layers and Pre-LN to the deeper layers. The rationale behind this is that Post-LN enhances gradient flow in the deeper layers, while Pre-LN stabilizes gradients in the earlier layers. By employing Post-LN in the initial layers and Pre-LN

in the later layers, `Mix-LN` promotes healthier gradient norms in the middle and deeper layers, fostering more balanced training across the entire network and ultimately improving the model's overall performance.

Our extensive experiments, spanning models from `70M` to `7B` parameters, demonstrate that `Mix-LN` consistently outperforms Pre-LN, Post-LN, and their variants. `Mix-LN` not only avoids the training instability associated with Post-LN but also significantly improves the quality of deeper layers compared to Pre-LN, leading to better pre-training performance. Additionally, models pre-trained with `Mix-LN` demonstrate superior learning during Supervised Fine-Tuning (SFT) and Reinforcement Learning from Human Feedback (RLHF) compared to those trained with Pre-LN or Post-LN, underscoring the importance of high-quality deep layers in LLMs.

## 2 HYPOTHESIS EVALUATION

In this section, we will evaluate our hypothesis that the inefficiency of deeper layers in LLMs stems from the choice of Pre-LN. The evaluation details are described as follows.

### 2.1 LAYER NORMALIZATION AND ITS GRADIENT

Figure 1 (a) and (b) illustrate Post-LN and Pre-LN Transformer architectures, respectively. Formally, let us define $x$ as the input, $\mathcal{F}(x)$ as either a FFN layer or a multi-head attention layer, and $\text{LN}(\cdot)$ as the layer normalization. Post-LN applies $\text{LN}(\cdot)$ after the residual addition:

$$\text{Post-LN}(x) = \text{LN}(x + \mathcal{F}(x)). \tag{1}$$

In contrast, Pre-LN applies $\text{LN}(\cdot)$ before the residual addition:

$$\text{Pre-LN}(x) = x + \mathcal{F}(\text{LN}(x)). \tag{2}$$

We can calculate the derivatives of Equations (1) and (2), as follows:

$$\frac{\partial \text{Post-LN}(x)}{\partial x} = \frac{\partial \text{LN}(x + \mathcal{F}(x))}{\partial (x + \mathcal{F}(x))} \left( I + \frac{\partial \mathcal{F}(x)}{\partial x} \right), \tag{3}$$

$$\frac{\partial \text{Pre-LN}(x)}{\partial x} = I + \frac{\partial \mathcal{F}(\text{LN}(x))}{\partial \text{LN}(x)} \frac{\partial \text{LN}(x)}{\partial x}. \tag{4}$$

Both the above equations involve an important component, i.g., the Jacobian matrix of layer normalization, $\mathbf{J}_{LN}(x') = \frac{\partial \text{LN}(x')}{\partial x'}$, where $x'$ is the input of $\text{LN}(\cdot)$. Following the proof of Xiong et al. (2020); Takase et al. (2023) with the assumption that $x'$ follow a normal distribution with a mean of 0, we can have:

$$\frac{\partial \text{LN}(x')}{\partial x'} = \frac{\sqrt{d}}{\|x'\|_2} \left( I - \frac{x'x'^{\top}}{\|x'\|_2^2} \right) \tag{5}$$

where $\sigma_{x'}$ is the standard deviations of $x'$ and $d$ is the hidden dimention. Hence,

$$\frac{\partial \text{LN}(x')}{\partial x'} = \frac{\sqrt{d}}{\sigma_{x'}\sqrt{d}} \left( I - \frac{x'x'^{\top}}{\sigma_{x'}^2 d} \right) = \frac{1}{\sigma_{x'}} \left( I - \frac{zz^{\top}}{d} \right). \tag{6}$$

where $z = (x' - \mu_{x'})/\sigma_{x'}$ is the standard normal distribution obtained after layer normalization. Since $d \gg 1$ in LLMs, we can finally obtain:

$$\frac{\partial \text{LN}(x')}{\partial x'} = \frac{1}{\sigma_{x'}} I. \tag{7}$$

In practice, we observe that $\sigma_{x'}$ gradually grows larger than one during training, which means the spectral norm of the Jacobian matrix of LN is smaller than 1. According to the derivative of Post-LN in Equation (3), this down-scaling factor will accumulate as $\prod_{l=1}^{L} \frac{1}{\sigma_{x'}^l}$ over multiple layers $L$, leading to gradient vanishing in early layers in Post-LN Transformers.

In contrast, the derivative of the residual connection in Pre-LN is decoupled from the term associated with the derivative of LN, as shown in Equation (4). This design helps prevent the vanishing gradient problem in early layers. However, since Pre-LN does not constrain the residual connection, the outputs of successive transformer blocks accumulate as the layer depth grows. Consequently, the derivative of Pre-LN in Equation (4) approaches an identity matrix, indicating that the entire Pre-LN operation of Equation (4) ceases to contribute effectively to learning. This explains why deeper layers in Pre-LN tend to contribute less to the model's overall learning compared to earlier layers.

## 2.2 EMPIRICAL EVALUATION SETUP

**Methods:** Our evaluation methodology involves a comparative analysis of two models—one utilizing Pre-LN and the other employing Post-LN. By empirically assessing the effectiveness of layers across different depths in each model, we expect to see that Pre-LN models will exhibit a decrease in the effectiveness of deeper layers, whereas Post-LN models will show sustained or even improved quality in deeper layers.

**LLM Models:** To rigorously evaluate our hypothesis, we conduct experiments on two distinct categories of LLMs: (i) *Open-weight large-scale LLMs* and (ii) *In-house small-scale LLMs*. In the open-weight category, we select LLaMa2-7B (Touvron et al., 2023) as a representative Pre-LN model and BERT-large (Devlin, 2018) as a Post-LN model. However, these open-weight models differ not only in normalization but also in other factors such as training data, activation functions, and context length, complicating our ability to isolate the impact of normalization alone. To control for these confounding variables, we conduct a second set of experiments by training small-scale LLMs from scratch ourselves. The goal is to ensure that the only difference between the models is the choice of layer normalization. Specifically, we train LLaMa-130M models on the C4 dataset with either Pre-LN or Post-LN, using RMSNorm (Zhang & Sennrich, 2019) and SwiGLU activations (Shazeer, 2020), following Lialin et al. (2023b); Zhao et al. (2024). Please refer to Appendix A for more training configuration details.

**Evaluation Metrics:** A critical challenge in validating our hypothesis lies in defining and selecting robust metrics that capture the effectiveness of individual layers. In this study, we employ two metrics: (i) *Angular Distance* and (ii) *Performance Drop*, which provide a meaningful evaluation of the role and contribution of each layer. In addition, we report the *gradient norm* of each layer to demonstrate the effect of different layer normalization on the gradient flow.

(i) *Angular Distance* $d(x^\ell, x^{\ell+n})$ is used in Gromov et al. (2024) to measure the angular distance between the input to layer $\ell$ and the input to layer $\ell + n$ on a neutral pre-training dataset. Formally, assuming $x_T^\ell$ is the input to the layer $\ell$, and $x_T^{\ell+n}$ is the input to the layer $\ell + n$, the angular distance between layers $\ell$ and and its subsequent $n^{th}$ layer, i.e., $\ell + n$, on a single token $T$ is given by

$$d(x^\ell, x^{\ell+n}) = \frac{1}{\pi} \arccos \left( \frac{x_T^\ell \cdot x_T^{\ell+n}}{\|x_T^\ell\| \|x_T^{\ell+n}\|} \right) \tag{8}$$

where $\| \cdot \|$ denotes the $L^2$-norm, and the factor of $1/\pi$ scales $d(x^\ell, x^{\ell+n})$ to the range [0, 1]. To eliminate the effect of randomness, the angular distance reported in this paper is averaged over 256K tokens from the C4 dataset. A smaller value of $d(x^\ell, x^{\ell+n})$ indicates a shorter distance, meaning that the two vectors are more similar. Layers whose representations are extremely similar to their neighboring layers mean that they can be easily removed, and therefore their weights are less effective. Ideally, representation should change substantially from layer to layer in order to most effectively make use of the parameters of a network (Yang et al., 2023; Gromov et al., 2024).

(ii) *Performance Drop* $\Delta P^{(\ell)}$ refers to the difference in the performance of an LLM before and after pruning the layer $\ell$. It quantifies the performance degradation caused by the removal of that layer. Formally, it can be defined as follows:

$$\Delta P^{(\ell)} = P_{\text{pruned}}^{(\ell)} - P_{\text{original}} \tag{9}$$

where $P_{\text{original}}$ is the performance of the model without any pruning, $P_{\text{pruned}}^{(\ell)}$ is the performance of the model after pruning layer $\ell$. A smaller value of $\Delta P^{(\ell)}$ indicates that removing the layer causes minimal change to the model's output, suggesting the layer is less important. Specifically, for LLaMA2-7B, we choose the commonly used MMLU (Hendrycks et al., 2020) as the evaluation task; for BERT-large, we opt for SQuAD v1.1 (Rajpurkar, 2016) as the evaluation task. Given the limited capacity of our in-house trained LLMs, we choose ARC-e (Clark et al., 2018) after supervised fine-tuning, instead of MMLU, for performance drop.

## 2.3 EVALUATION RESULTS

### 2.3.1 OPEN-WEIGHT LARGE-SCALE LLMS

Figure 2-(a, c) illustrate the metric values for BERT-Large. Both metrics indicate that, as a Post-LN model, the early layers of BERT-Large are less effective compared to the deeper layers. As shown in

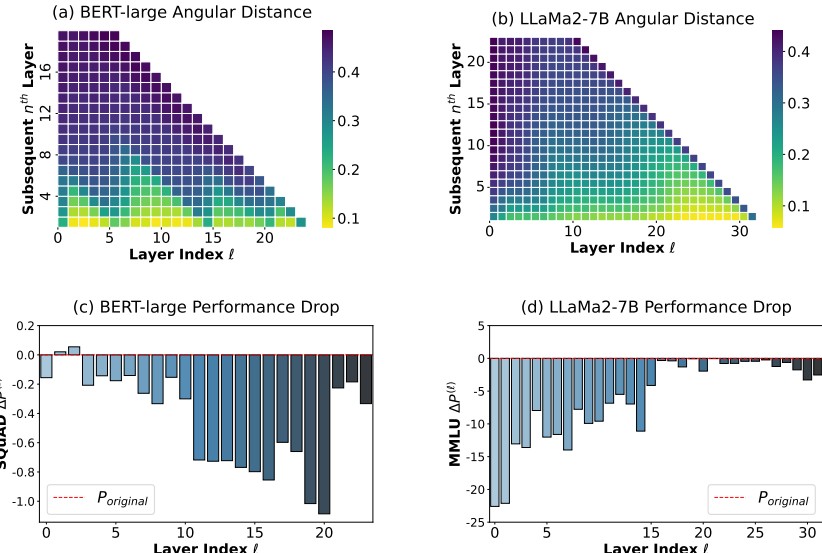

Figure 2: Results of open-weight large-scale LLMs. **Angular Distance (a, b)**: Each column represents the angular distance from the initial layer $\ell$ (x-axis) and its subsequent $n^{th}$ layer (y-axis). The distance is scaled to the range [0, 1], where yellow indicates smaller distances and purple indicates larger distances. **Performance Drop (c, d)**: (c): SQuAD v1.1 performance drop of removing each layer from BERT-large; (d): MMLU accuracy drop of removing each layer from LLaMa2-7B.

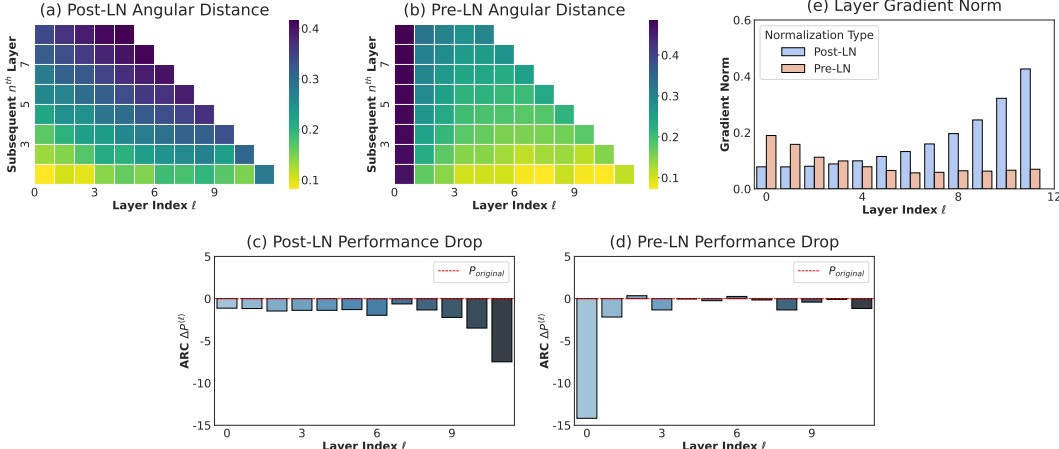

Figure 3: Results of in-house small-scale LLaMa-130M. **Angular Distance (a, b)**: Each column represents the angular distance from the initial layer $\ell$ (x-axis) and its subsequent $n^{th}$ layer (y-axis). The distance is scaled to the range [0, 1], where yellow indicates smaller distances and purple indicates larger distances. **Performance Drop (c, d)**: ARC-e performance drop of removing each single layer from LLaMa-130M. **Gradient Norm (e)**: Gradient norm of each layer in LLaMa-130M.

Figure 2-a, the first half of BERT-Large tends to have a smaller angular distance (more yellow) from neighboring layers than the second half. In particular, layers 3, 4, 9, 10, and 11 show a very high similarity to their subsequent layers. In Figure 2-c, the performance drop on SQuAD of removing an early layer is significantly smaller than the impact of removing a deeper layer. Intriguingly, removing layers 2 and 3 can even improve the performance slightly.

In contrast, Figure 2-(b, d) display the metric values for LLaMa2-7B. As a Pre-LN model, the angular distance between neighboring layers decreases gradually (from purple to yellow) from the top layers to the 30th layer as illustrated in Figure 2-b. Notably, the deeper layers (20th to 30th) exhibit extremely small angular distances to their adjacent layers. This trend is consistent with the

MMLU performance in Figure 2-d, where the removal of deeper layers results in almost negligible accuracy loss while removing early layers causes a substantial drop in accuracy.

In summary, we observe that the least effective layers in LLaMa2-7B are located in the deeper layers, whereas the early layers in BERT-Large are less effective than the deeper layers. The results from the category of open-weight large-scale LLMs strongly support our hypothesis, demonstrating a clear alignment with our expectations.

### 2.3.2 IN-HOUSE SMALL-SCALE LLMs

Figure 3 illustrates all metric values for two LLaMa-130M models, where the only difference between them is the choice of layer normalization.

Figures 3-(a, b) show the Angular Distance for Post-LN and Pre-LN, respectively. Without the effects of other compounding factors, this comparison provides a clearer distinction between Post-LN and Pre-LN compared to open-weight large-scale LLMs. In Post-LN models, the most similar layers are concentrated in the early stages, with the first three layers showing particularly low distance. As the depth increases, the layers become increasingly distinctive. In contrast, the Pre-LN LLaMa-130M exhibits a gradual decrease in angular distance as depth increases, leading to highly similar deep layers. Figures 3-(d, e) further confirm this with the Performance Drop metric: removing early layers (e.g., 0-7 layers) in Post-LN results in minimal performance loss, while deeper layers (especially layers 9-11) are critical to preserving the original performance. However, Pre-LN LLaMa-130M exhibits the opposite trend, where removing most layers after the first layer causes negligible performance loss, indicating that they contribute little to the model's output.

Figure 3-(c) shows the gradient norm of each layer for Post-LN and Pre-LN at the beginning of the training. The results perfectly align with our expectations: Post-LN leads to larger gradients in deeper layers but suffers from severe gradient vanishing in early layers, whereas Pre-LN maintains healthy gradient flow in early layers but diminishes in later layers.

With the consistent findings from both open-weight LLMs and our in-house LLMs, we can conclude that the widespread use of Pre-LN in LLMs is the root cause of the ineffectiveness of deep layers.

## 3 MIX-LAYER NORMALIZATION (MIX-LN)

Having validated our hypothesis that the use of Pre-LN is the root cause of the ineffectiveness of deeper layers, we propose **Mix-Layer Normalization (Mix-LN)**, a novel normalization strategy designed to enhance the effectiveness of both middle and deeper layers in LLMs.

The key idea behind Mix-LN is to leverage the strengths of both Pre-LN and Post-LN. Post-LN has been shown to improve the effectiveness of deeper layers, while Pre-LN is more effective for earlier layers. Therefore, we propose to apply Post-LN to the initial layers and Pre-LN to the later layers, ensuring that the middle and deeper layers benefit from the advantages of both methods.

Formally, for an LLM with $L$ layers, we apply Post-LN to the first $\lfloor aL \rfloor$ layers and Pre-LN to the remaining $\lceil (1-a)L \rceil$ layers, where $a \in [0,1]$ is a hyperparameter controlling the transition point between the two normalization strategies. The functions $\lfloor \cdot \rfloor$ and $\lceil \cdot \rceil$ denote the floor and ceiling operations, respectively. Although the final layers may still experience smaller gradients due to the use of Pre-LN, the negative impact is substantially mitigated because the number of layers employing Pre-LN is reduced from $L$ to $\lceil (1-a)L \rceil$. This reduction improves gradient flow in the deeper layers, enhancing their effectiveness. Additionally, we expect that Mix-LN can alleviate training instability issues caused by Post-LN (Nguyen & Salazar, 2019; Wang et al., 2024), as reducing the number of layers using Post-LN leads to a smaller accumulation of gradient attenuation, according to the analysis in Section 2.1.

## 4 MAIN EXPERIMENTAL RESULTS

### 4.1 LLM PRE-TRAINING

In this section, we verify the effectiveness of Mix-LN by comparing it with various common normalization techniques, including Post-LN (Nguyen & Salazar, 2019), DeepNorm (Wang et al.,

2024), and Pre-LN (Dai, 2019). Following Lialin et al. (2023a); Zhao et al. (2024), we conduct experiments using the LLaMA-based architecture with various sizes from 71M to 1B parameters, incorporating RMSNorm (Shazeer, 2020) and SwiGLU activations (Zhang & Sennrich, 2019). Models are trained with Adam (Kingma, 2014) using different learning rates based on model size: specifically, we use a learning rate of 1e-3 for models with 250M parameters and below, and a learning rate of 5e-4 for the 1B parameter model. All models of the same size are trained with identical configurations except for the normalization. To determine the optimal value for the hyperparameter $\alpha$ in `Mix-LN`, we performed a small hyperparameter sweep using LLaMA-250M, as shown in Table 5. We found that $\alpha = 0.25$ provided the best performance, and therefore, we applied this value across all model sizes.

Table 1: Perplexity ($\downarrow$) comparison of various normalization methods across various LLaMA sizes.

|  | LLaMA-71M | LLaMA-130M | LLaMA-250M | LLaMA-1B |
| --- | --- | --- | --- | --- |
| Training Tokens | 1.1B | 2.2B | 3.9B | 5B |
| Post-LN | 35.18 | 26.95 | 1409.09 | 1411.54 |
| DeepNorm | 34.87 | 27.17 | 22.77 | 1410.94 |
| Pre-LN | 34.77 | 26.78 | 21.92 | 18.65 |
| Mix-LN | **33.12** | **26.07** | **21.39** | **18.18** |

Results are shown in Table 1. Post-LN generally yields the worst performance and even diverges with larger models, aligning with previous studies that indicate Post-LN suffers from training instability in Transformers (Xiong et al., 2020; Takase et al., 2022). DeepNorm, as a modified version of Post-LN, achieves comparable performance to Pre-LN with smaller model sizes; however, it also experiences divergence during training with 1B parameter models. This observation confirms severe training instability of Post-LN, where gradients in early layers vanish, preventing proper model convergence. In contrast, `Mix-LN` consistently achieves the lowest perplexity across various model sizes. `Mix-LN` achieves a notable gain by 1.65 and 0.53 perplexity with LLaMA-71M and LLaMA-250M, respectively, compared to the widespread Pre-LN.

The above results clearly show that `Mix-LN` not only overcomes the instability of Post-LN but also enhances the model quality by combining the benefits of Pre-LN and Post-LN, making it an ideal choice for large-scale LLMs.

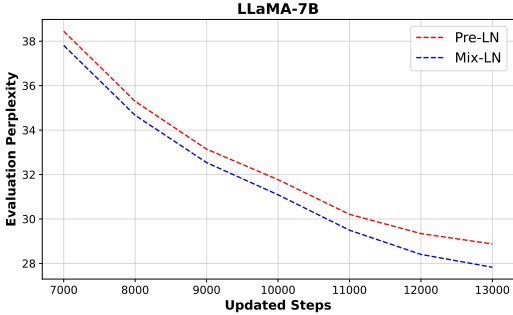

Figure 4: Training curve (eval perplexity) of `Mix-LN` and Pre-LN with LLaMa-7B.

## 4.2 SCALING UP TO 7B MODEL

It is essential to evaluate if the benefits of `Mix-LN` also hold for larger model sizes. To this end, we conducted experiments using the LLaMa-7B architecture, which features an embedding size of 4096 and 32 total layers. All training configurations were kept identical, except for the layer normalization method. Due to computational constraints, we managed to complete only 13,000 steps of training. The training curve is presented in Figure 4. We can see that it is possible to scale the gain of `Mix-LN` to larger-scale training.

Nevertheless, we found that `Mix-LN` becomes more sensitive to $\alpha$ when scaling up to 7B training. In general, smaller values of $\alpha$ and a longer warm-up period are required to stabilize the training of `Mix-LN` at a larger scale. For instance, we use a relatively small $\alpha = 6.25\%$ here to ensure stable training of LLaMa-7B.

## 4.3 SUPERVISED FINE-TUNING

Table 2: Fine-tuning performance (↑) of LLaMa with various normalizations.

| Method | MMLU | BoolQ | ARC-e | PIQA | Hellaswag | OBQA | Winogrande | Avg. |
|--------|------|-------|-------|------|-----------|------|------------|------|
| **LLaMA-250M** | | | | | | | | |
| Post-LN | 22.95 | 37.83 | 26.94 | 52.72 | 26.17 | 11.60 | 49.56 | 32.54 |
| DeepNorm | 23.60 | 37.86 | 36.62 | 61.10 | 25.69 | 15.00 | 49.57 | 35.63 |
| Pre-LN | 24.93 | 38.35 | 40.15 | 63.55 | 26.34 | 16.20 | 49.01 | 36.93 |
| Mix-LN | **26.53** | **56.12** | **41.68** | **66.34** | **30.16** | **18.00** | **50.56** | **41.34** |
| **LLaMA-1B** | | | | | | | | |
| Post-LN | 22.95 | 37.82 | 25.08 | 49.51 | 25.04 | 13.80 | 49.57 | 31.96 |
| DeepNorm | 23.35 | 37.83 | 27.06 | 52.94 | 26.19 | 11.80 | 49.49 | 32.67 |
| Pre-LN | 26.54 | **62.20** | 45.70 | 67.79 | 30.96 | 17.40 | 50.51 | 43.01 |
| Mix-LN | **27.99** | 61.93 | **48.11** | **68.50** | **31.35** | **18.80** | **55.93** | **44.66** |

We believe that the superior middle and deeper layers produced by Mix-LN are better equipped to learn during supervised fine-tuning. This advantage stems from the fact that these layers capture more diverse and rich features compared to those trained with Pre-LN. In complex downstream tasks, having access to a broad spectrum of features allows the model to make more nuanced predictions, leading to improved generalization.

To verify our conjecture, we follow Li et al. (2024) and fine-tune the models obtained in Section 4.1 on Commonsense170K (Hu et al., 2023), evaluating them on eight downstream tasks. As shown in Table 2, Mix-LN consistently outperforms other normalization techniques across all evaluated datasets. For the LLaMA-250M model, Mix-LN achieves a significant average gain of 4.26% and a 17.31% improvement on BoolQ compared to Pre-LN. Similar trends are observed with the larger LLaMA-1B model. Even though Mix-LN only slightly reduces perplexity by 0.25 compared to Pre-LN, it delivers substantial performance gains in supervised fine-tuning.

## 4.4 REINFORCEMENT LEARNING FROM HUMAN FEEDBACK

Table 3: RLHF comparison of final reward (↑) of Pre-LN and Mix-LN with LLaMA-1B.

| Method | Model | Final Reward |
|--------|-------|--------------|
| Pre-LN | LLaMA-1B | 0.75 |
| Mix-LN | LLaMA-1B | **1.32** |

Consistently, the benefits of Mix-LN can be seamlessly transferred to RLHF. Following Adam-mini (Zhang et al., 2024), we implement the RLHF workflow from InstructGPT (Ouyang et al., 2022) and train 1B models obtained in Section 4.1 on the ultrafeedback dataset to optimize the preference reward. Table 3 illustrates that Mix-LN achieves a notable reward gain (higher is better) over Pre-LN, i.e., 1.32 vs. 0.75.

## 4.5 EVALUATION WITH VISION TRANSFORMERS

Table 4: Accuracy (↑) comparison of Pre-LN and Mix-LN on ViT models.

| Model | ViT-Tiny | ViT-Small |
|-------|----------|-----------|
| Pre-LN | 67.30 | 75.99 |
| Mix-LN | **67.34** | **76.40** |

To evaluate Mix-LN on non-language models, we replace Pre-LN in ViT models with Mix-LN with $\alpha = 0.25$. We train the updated model for 120 epochs on ImageNet-1K following Liu et al. (2022a), using the ConvNeXt (Liu et al., 2022b) configurations. The results clearly demonstrate that the benefits of Mix-LN also generalize to non-language models. The results in Table 4 demonstrate that the benefits of Mix-LN extend to non-language tasks, with performance gains that are more pronounced in larger models (ViT-Small) compared to smaller ones (ViT-Tiny).

## 5 ANALYSIS AND MORE EVALUATIONS

Table 5: Perplexity of LLaMA-1B with various Post-LN ratios $\alpha$.

|  | Pre-LN | Mix-LN | | | | | Post-LN |
|---|---|---|---|---|---|---|---|
| Post-LN ratios $\alpha$ | 0 | 16.7% | 25.0% | 33.0% | 41.7% | 50.0% | 100% |
| Perplexity | 18.65 | 18.34 | **18.18** | 18.41 | 18.55 | 18.86 | 1434 |

**What is the proper Post-LN ratio $\alpha$ for `Mix-LN`?** `Mix-LN` has a hyperparameter, $\alpha$, that controls the ratio of layers applying Post-LN. Specifically, $\alpha = 0$ means Pre-LN is applied to all layers, while $\alpha = 1$ corresponds to pure Post-LN. To determine the optimal Post-LN ratio, we conduct a sweep over the values [0, 16.7%, 25.0%, 33.0%, 41.7%, 50.0%, 100%] using LLaMA-1B on the C4 dataset. The results are shown in Table 5. As the normalization transitions from Pre-LN to Mix-LN, the model achieves progressively lower perplexity, reaching its best performance at $\alpha = 0.25$. Beyond this point, performance begins to decline, although it still surpasses that of pure Pre-LN until most layers apply Post-LN, where performance degrades significantly. Based on these results, we choose $\alpha = 0.25$ for all model sizes, although we believe there is potential to further improve the performance of `Mix-LN` by searching for the optimal $\alpha$ for each individual model.

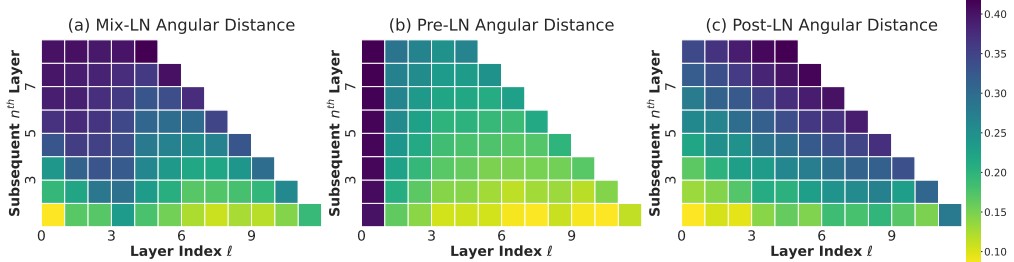

Figure 5: Angular distance from initial layer $\ell$ (x-axis) with block size $n$ (y-axis) of LLaMA-130M.

**`Mix-LN` promotes representation diversity across layers.** As we have claimed, our hybrid approach promotes a more balanced gradient flow throughout the entire network. To validate this, we report the angular distance of LLaMA-130M for Pre-LN, Post-LN, and `Mix-LN` in Figure 5. Given block size $n$, the layers with the smallest distances are highlighted in the lightest yellow in each row. Notably, `Mix-LN` consistently exhibits larger distances (darker color) across layers compared to Pre-LN, except for the final two layers. This indicates that `Mix-LN` produces more diverse representations between layers than Pre-LN. In contrast, the smallest distances in Post-LN are concentrated in the early layers, reinforcing the notion that Post-LN tends to restrict representation diversity in deeper layers.

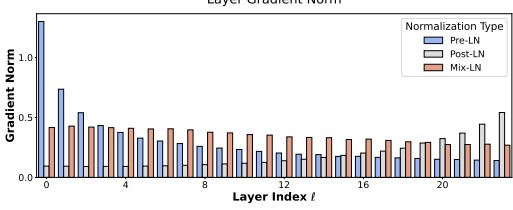
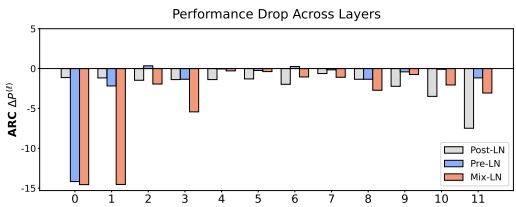

(a) Layer gradient norm of LLaMA-250M with various normalization techniques.

(b) Performance drop comparison of LLaMA-130M across layers for Pre-LN, Post-LN, and Mix-LN.

Figure 6: Comparison of gradient norms and performance drops of layer pruning for different layer normalization.

**`Mix-LN` enhances healthier gradient norms across all layers.** We compare the gradient norm of different LN at initialization in Figure 6a. It demonstrates that `Mix-LN` maintains more consistent

gradient norms across all layers. This balance results in a more uniform distribution of gradient norms across layers, allowing all parts of the network—both shallow and deep layers—to contribute effectively to model training.

**Performance drop of layer pruning for `Mix-LN`.** To further evaluate the effectiveness of Mix-LN, we compare the performance drop ($\Delta P$) across layers with Pre-LN and Post-LN. Fig. 6b shows Mix-LN achieves a more significant contribution from deeper layers. Specifically, the deeper layers in Mix-LN models show a larger $\Delta P$ compared to Pre-LN, indicating that these layers contribute more effectively to the model's overall performance.

**Comparing with other Layer Normalization.** In addition, we conducted comparisons using LLaMA-250M to evaluate Mix-LN against recently proposed normalization methods, including Admin (Liu et al., 2020), Sandwich-LN (Ding et al., 2021), and Group-LN (Wu & He, 2018; Ma et al., 2024). The results indicate that Sandwich-LN and Group-LN slightly outperform Pre-LN, while Admin performs worse. However, all of these methods fail to reduce perplexity below 23, falling short of Mix-LN. This result highlights the effectiveness of Mix-LN compared to other recent innovations.

Table 6: Comparison against other normalization methods on LLaMA-250M.

| Model | Pre-LN | Admin | Group-LN | Sandwich-LN | Mix-LN |
|-------|--------|-------|----------|-------------|--------|
| LLaMA-250M | 23.39 | 24.82 | 23.10 | 23.26 | **22.33** |

## 6 CONCLUSION

In this paper, we have addressed the inefficiencies of deep layers in LLMs by identifying the widespread use of Pre-LN as the root cause. Pre-LN leads to diminished gradients in deeper layers, reducing their effectiveness. While Post-LN preserves deeper gradients, it suffers from vanishing gradients in earlier layers. To resolve this, we introduced `Mix-LN`, a hybrid normalization technique that combines the strengths of both Pre-LN and Post-LN. By applying Post-LN to early layers and Pre-LN to deeper layers, `Mix-LN` achieves balanced gradient norms throughout the network, enabling more effective training. Our experiments show that `Mix-LN` consistently outperforms both Pre-LN and Post-LN, enhancing pre-training and fine-tuning performance without increasing model size. By fully utilizing the potential of deep layers, `Mix-LN` improves the overall capacity and efficiency of LLMs.

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

## A  DETAILS OF EXPERIMENTS

### A.1  ARCHITECTURE AND HYPERPARAMETERS

We introduce details of the LLaMA architecture and hyperparameters used for pre-training following (Lialin et al., 2023a; Zhao et al., 2024). Table 7 shows the most hyperparameters of LLaMA models across model sizes. We use a max sequence length of 256 for all models, with a batch size of 512, and a total of 131K tokens per batch. Learning rate warmup is applied to the first 10% of the training steps. We train models using Adam with a cosine annealing for the learning rate schedule, decaying to 10% of the initial learning rate. We use a learning rate of 1e-3 for models with 250M parameters and below, and a learning rate of 5e-4 for the 1B parameter model.

Table 7: Hyperparameters of LLaMA models used in this paper.

| Params | Hidden | Intermediate | Heads | Layers | Steps | Data amount | LR | Batch Size | $\alpha$ |
|---|---|---|---|---|---|---|---|---|---|
| 71M | 512 | 1368 | 8 | 12 | 10K | 1.1 B | $1 \times 10^{-3}$ | 512 | 25% |
| 130M | 768 | 2048 | 12 | 12 | 20K | 2.2 B | $1 \times 10^{-3}$ | 512 | 25% |
| 250M | 1024 | 2560 | 16 | 24 | 40K | 3.9 B | $1 \times 10^{-3}$ | 512 | 25% |
| 1 B | 2048 | 5461 | 32 | 24 | 100K | 5.0 B | $5 \times 10^{-4}$ | 512 | 25% |
| 7 B | 4096 | 11008 | 32 | 32 | 13K | 1.7 B | $5 \times 10^{-4}$ | 512 | 6.25% |

## B  COMPATIBILITY TO ADVANCED

In this section, we also evaluate if `Mix-LN` can integrate well with the advanced techniques proposed to stabilize training. Specifically, we evaluate the commonly used Scaled Initialization (Nguyen & Salazar, 2019; Scao et al., 2022) that initializes $W_2$ and $W_0$ with a smaller normal distribution $\mathcal{N}(0, \sqrt{2/5d}/\sqrt{2N})$ to stabilize training dynamics; and Scaled Embed (Takase et al., 2023) scales up embeddings to stabilize LayerNorm gradients. We observe that both Pre-LN and `Mix-LN` work effectively with Scaled Initialization. However, incorporating Scaled Embed on top of this setup leads to a degradation in performance.

Table 8: Perplexity of LLaMA-130M with various normalization methods with Scaled Initialization and Scaled Embed.

| Normalization | Scaled Initialization | Scaled Embed | Perplexity |
|---|---|---|---|
| Pre-LN | | | 32.18 |
| Mix-LN | | | 29.95 |
| Pre-LN | ✓ | | 30.63 |
| Mix-LN | ✓ | | 29.77 |
| Pre-LN | ✓ | ✓ | 31.28 |
| Mix-LN | ✓ | ✓ | 31.19 |

## C  RELATED WORK

### C.1  NORMALIZATION IN LANGUAGE MODELS

Layer Normalization (LN), first proposed by Ba (2016), has become the de facto standard for normalizing activations in modern language models. It directly estimates normalization statistics from the summed inputs to neurons within a hidden layer, ensuring that the input distribution to each layer remains stable throughout training. In the original Transformer (Vaswani, 2017), LN was initially applied after the residual connection, a configuration known as Post-LN. However, subsequent studies (Baevski & Auli, 2018; Dai, 2019; Nguyen & Salazar, 2019) found that placing LayerNorm before the residual connection (Pre-LN) results in more stable performance, especially in large language models (Brown, 2020; Touvron et al., 2023; Jiang et al., 2023). Xiong et al. (2020) theoretically demonstrated that Post-LN results in larger gradients near the output layer, making the use of

warm-up essential to avoid instability is necessary. Conversely, Pre-LN scales down gradients with the depth of the model, which ensures more stable gradients during initialization. Our work builds upon Xiong et al. (2020), highlighting that while Pre-LN prevents instability by reducing gradient magnitudes, smaller gradients in deeper layers can diminish the effectiveness of the corresponding weights.

To improve the effectiveness of deeper layers in language models, various LN variants have been proposed. For instance, Wang et al. (2019) verified empirically that Post-LN suffers from gradient vanishing in deep Transformers, while Pre-LN facilitates stacking more layers. They consequently introduced dynamic linear combination of layers (DLCL), which connects all previous layers to improve trainability. Similar techniques have been employed in other works (Bapna et al., 2018; Dou et al., 2018). Liu et al. (2020) revealed that Post-LN has strong dependencies on the residual branch, often leading to instability. To address this, Adaptive Model Initialization (Admin) was introduced, which uses additional parameters to control residual dependencies in Post-LN, stabilizing training. DeepNorm (Wang et al., 2024) further improved the trainability of deep Transformers by upscaling the residual connection before applying LN, reducing model updates, and enabling deeper architectures. Additionally, Ding et al. (2021) proposed Sandwich LayerNorm, normalizing both the input and output of each transformer sub-layer. Takase et al. (2022) identified that Post-LN tends to preserve larger gradient norms in deeper layers, potentially leading to more effective training. To address the issue of gradient vanishing in early layers, they introduced B2T, a method that uses a residual connection to bypass all LN except the final one in each layer. Huang et al. (2025) unveiled that LN layers suffer from loss spikes and gradient spikes during LLM training. We got inspiration from Takase et al. (2022), addressing the limitations of both Pre-LN and Post-LN by combining them. We study Scaled Initialization and Scaled Embed in Appendix B.

## C.2 INEFFICACY OF DEEP LAYERS IN LLMS

The Inefficacy of deep layers in LLMs serves as a valid indicator for LLM pruning. Yin et al. (2023) demonstrated that the deeper layers of prominent LLMs like LLaMA and Mistral can be pruned more aggressively than earlier layers, without causing a significant drop in performance. Similarly, Gromov et al. (2024) and Men et al. (2024) further explored layer pruning, identifying the deeper layers of LLMs as typically less essential. Lad et al. (2024) observed that in models like Pythia and GPT-2, deeper layers exhibit strong resilience to interventions, such as layer deletion or swapping. Our work shares similarities with Gromov et al. (2024) in applying angular distance to assess the effectiveness of layers. However, while they identify the inefficacy of deeper layers, they do not offer an explanation for this phenomenon nor propose a solution to address it. Building on the insights from this paper, Sun et al. (2025) introduced the concept of the "Curse of Depth" to highlight the pitfalls of deeper layers in LLMs. To address this issue while avoiding the instability associated with Post-LN, they proposed LayerNorm Scaling to scale down the output of LN by its depth.

