# OpenReview forum: "Mix-LN: Unleashing the Power of Deeper Layers by Combining Pre-LN and Post-LN"
_ICLR.cc/2025/Conference — ICLR 2025 Poster_

### Official Review · Reviewer_mk56 · 2024-10-30

**Soundness:** 3
**Presentation:** 3
**Contribution:** 2
**Rating:** 6
**Confidence:** 4

**Summary:**

This paper identifies an training shortfall of LLMs that their deep layers contribute minimally to the overall performance, which wastes the model capacity. The paper identifies the reason behind this as the use of Pre-LN leading to diminished gradients in later layers. The paper further proposes to use a mixed LN approach, where post-LNs are applied to the earlier layers and pre-LNs to the later layers. Experiments show enhancement brought by the mixed LN approach in both pretraining and supervised finetuning.

**Strengths:**

This paper provides novel and interesting insights on the different impact of pre/post LN in the training behavior of earlier and later layers. The observation leads to a straightforward solution of applying mixed LN, which can be easily applied in all models. The paper is well-written and easy to follow. Experiments on multiple models and tasks, as well as ablation studies show solid performance improvement of the proposed method.

**Weaknesses:**

One major weakness of this paper is that the provided results are not adequate to support the claim that mixed-LN helps deep model by allowing both shallow and deep layers contributes effectively. From the main results in Tab. 1 and 2, it appears that mix-LN improves more over pre-LN in that smaller 71M model (12 layers) than the deeper 1B model (32 layers), which seems to suggest the benefit of mix LN is less in deep models. It would be better to provide results of even larger and deeper models to show that the benefit of mix LN do scale up. If computation resource is limiting the further experiments on larger models, trying a slim but deep architecture may be helpful to verify the effectiveness of the proposed method.

**Questions:**

See weakness part. I would like to see more evidence that the proposed method helps more on deeper models.

---

> ### Author Response · Authors · 2024-11-20
>
> ### **Response to Reviewer mk56**
> We sincerely thank the reviewer for giving us constructive review. We thank the reviwer for recognizing our paper is novel and provides interesting insights. We address your concerns in the following.
>
> **W1: One major weakness of this paper is that the provided results are not adequate to support the claim that mixed-LN helps deep model by allowing both shallow and deep layers contributes effectively. It would be better to provide results of even larger and deeper models to show that the benefit of mix LN do scale up.**
>
> - First, we would like to clarify that the small performance gain reported in our submission of 1B model is due to a suboptimal choice of α. Specifically, for the 1B model, the ratio α was set as 0.33, rather than the optimal ratio of 0.25 as stated in our claims. When α is set to the optimal value of 0.25, Mix-LN achieves a notable performance improvement over Pre-LN, with a nearly 0.5 perplexity reduction for LLaMa-1B as demonstrated in the following table. Note that it is reasonable for the performance gains to reduced to certain extends as the model size increases, since larger models inherently have better baseline performance, reducing the room for further improvement.
>
>
>     | LLaMa-1B | Pre-LN | Mix-LN (α=0.33) | Mix-LN (α=0.25) |
>     | :---- | :---- | :---- | :---- |
>     | PPL | 18.65 | 18.40 | 18.18 |
>
> - More importantly, the improvement in perplexity translates into significant enhancements in downstream tasks, including self-supervised fine-tuning (SFT) and reinforcement learning from human feedback (RLHF). For instance, after applying SFT with the trained 1B models on the Commonsense170K dataset, Mix-LN achieves an average accuracy improvement of 1.24%, with a particularly remarkable gain of 5.42% on the Winogrande task.
>
>     | Method | MMLU | BoolQ | ARC-e | PIQA | Hellaswag | OBQA | Winogrande | Avg. |
>     | :---- | :---- | :---- | :---- | :---- | :---- | :---- | :---- | :---- |
>     | Pre-LN | 26.54 | 62.20 | 45.70 | 67.79 | 30.96 | 17.40 | 50.51 | 43.01 |
>     | Mix-LN | 27.99 | 61.93 | 48.11 | 68.50 | 31.35 | 18.80 | 55.93 | 44.25 |
>
> - Consistently, the benefits of Mix-LN can be seamlessly transferred to RLHF. Following Adam-mini (https://arxiv.org/pdf/2406.16793), we use the ultrafeedback dataset and implement the RLHF workflow from InstructGPT (https://arxiv.org/abs/2203.02155) to optimize the preference reward. Mix-LN achieves a notable reward gain (higher is better) over Pre-LN, i.e., 1.32 vs 0.75.
>
>     | Method | Model | Final Reward |
>     | :---- | :---- |:---- |
>     | Pre-LN |  LLaMA-1B | 0.75 |
>     | Mix-LN | LLaMA-1B | 1.32 |
>
> - To further address your concerns regarding the benefits of Mix-LN on large-scale models, we conducted experiments with **LLaMa-7B** using the same setup as GaLore. All training configurations were kept identical, except for the choice of layer normalization. Given the limited rebuttal time window and our computing resources, we only completed 13,000 steps of training. However, based on our experience, models that exhibit consistent improvements early in training typically retain these advantages through the later stages. The perplexity of Mix-LN and Pre-LN under these conditions is summarized in the table below. We also report the training curves of Mix-LN and Pre-LN in Figure 4 of our revision.
>
>     **Table: Perplexity of LLaMA-7B at Different Training Steps**
>
>     | Training Steps | 1000    | 2000 | 5000 | 8000  | 11000 | **13000** |
>     |----------------|---------|------|------|-------|-------|-------|
>     | Pre-LN         | 817.84 | 290.04 | 52.88 | 35.30 | 30.21 | 28.90 |
>     | Mix-LN         | 939.54 | 533.79 | 52.72 | 34.67 | 29.49 | **27.84** |

---

> > ### Comment · Reviewer_mk56 · 2024-11-20
> > **Sensitivity to alpha**
> >
> > I would like to thank the author for the response. The provided results largely resolves my concern on the scalability of the proposed method. However, it echoes a point raised by other reviewers, that the benefit of mix-LN may be largely dependent on the choice of $\alpha$. Can you comment on how to effectively choosing a proper $\alpha$ for a new model besides naively trying out each options?

---

> > > ### Author Response · Authors · 2024-11-20
> > > **Thank you for your quick response!**
> > >
> > > Thank you for your follow-up question and for acknowledging our response. We are glad that the provided results address your concerns regarding the scalability of our method.
> > >
> > > -  Regarding your point about the dependence of Mix-LN’s performance on the choice of $\alpha$, we agree that this is an important consideration, especially when applying Mix-LN to new models. From our experience, Mix-LN is quite robust to the choice of $\alpha$ as our results reported in the following tables using LLaMA-250M and LLaMA-1B. Results that outperform Pre-LN are highlighted in bold for clarity. We can see that as long as $\alpha$ is constrained below 0.5, Mix-LN consistently outperforms Pre-LN, with the best results achieved with $\alpha=0.25$.
> > >
> > >     **Table: Perplexity of LLaMA-250M with various Post-LN ratios $\alpha$**
> > >     | Post-LN Ratios ($\alpha$) | Pre-LN | Mix-LN (12.5%) | Mix-LN (25.0%) | Mix-LN (33.3%) | Mix-LN (41.7%) | Mix-LN (50.0%) | Mix-LN (75.0%) | Post-LN |
> > >     |---------------------|--------|----------------|----------------|----------------|----------------|----------------|----------------|---------|
> > >     | Perplexity      | 23.39  | **22.37**          | **22.33**      | **22.83**          | **22.80**          | **22.81**          | 23.64          | 32.18   |
> > >
> > >     ---
> > >
> > >     **Table: Perplexity of LLaMA-1B with various Post-LN ratios $\alpha$**
> > >     | Post-LN Ratios ($\alpha$)  | Pre-LN | Mix-LN (16.7%) | Mix-LN (25.0%) | Mix-LN (33.3%) | Mix-LN (41.7%) | Mix-LN (50.0%) | Post-LN |
> > >     |----------------|--------|--------|--------|--------|---------|---------|---------|
> > >     | Perplexity       | 18.65 | **18.34**  | **18.18**  | **18.41**  | **18.55**   | 18.86   | 1434    |
> > >
> > > -  Moreover, we also conducted experiments with ViT models on ImageNet-1K, where we directly set $\alpha=0.25$ without fine-tuning the hyperparameter. The results demonstrate that $\alpha=0.25$ consistently outperforms Pre-LN, with the performance gains being more pronounced for the larger model (ViT-Small) compared to the smaller model (ViT-Tiny).
> > >
> > >     |  | Pre-LN | Mix-LN |
> > >     | :---- | :---- | :---- |
> > >     | ViT-Tiny | 67.30 | **67.34** |
> > >     | ViT-Small | 75.99 | **76.40** |
> > >
> > > - Based on these observations, we believe that setting $\alpha=0.25$ is a robust choice that should yield good results in most cases. However, if $\alpha=0.25$ fails to converge for even larger models (although we haven’t tried and encountered such failure), we recommend reducing $\alpha$. This recommendation is informed by our observation that pure Post-LN leads to divergence with LLaMA-1B, suggesting that too many Post-LN layers may induce instability. Reducing $\alpha$ in such cases could help mitigate this issue.
> > >
> > > These heuristics are by no means exhaustive but can serve as a starting point. For future work, we plan to further investigate the optimal choice of $\alpha$ for different model sizes theoretically.
> > >
> > > We hope this addresses your concern, and we thank you again for the thoughtful question, which has inspired us to explore this direction further.

---

> > > > ### Comment · Reviewer_mk56 · 2024-11-27
> > > >
> > > > Thank you for the response. Though there are still some lingering issues like finding the optimal choice of alpha, I do believe this work provides interesting observation on the use of post/pre LN and provides a practical method to mix them. I will increase my score to acceptance.

---

> > > > > ### Author Response · Authors · 2024-11-27
> > > > > **Thank you for your support!**
> > > > >
> > > > > Dear Reviewer mk56,
> > > > >
> > > > > We sincerely appreciate your recognition and acceptance of our paper. It means a lot to us, and we are truly grateful for the time and effort you dedicated to reviewing our work. Your constructive feedback has been invaluable in helping us refine our research.
> > > > >
> > > > > Thank you once again for your thoughtful evaluation and support!
> > > > >
> > > > > Best wishes,
> > > > >
> > > > > Authors

---

> ### Author Response · Authors · 2024-11-25
>
> Dear Reviewer mk56,
>
> We sincerely appreciate your thoughtful feedback and follow-up questions, which indeed greatly improved our work. As we approach the conclusion of the discussion phase, please feel free to share any additional concerns, we would be more than happy to address them.
>
> Kind regards,
>
> The Authors

---

> ### Author Response · Authors · 2024-11-27
> **Kind reminder of our response**
>
> Dear Reviewer mk56,
>
> Thank you once again for your valuable feedback and for taking the time to review our work. We greatly appreciate the insights and constructive comments you have provided, which have helped us improve the clarity and quality of our paper.
>
> We would like to kindly ask if our response has satisfactorily addressed all of your concerns. If there are any remaining questions or points of clarification, we would be more than happy to provide further details.
>
> If you find that your concerns have been fully resolved, we would sincerely appreciate it if you might consider re-evaluating your score in light of the updated information.
>
> Thank you again for your time and thoughtful review. Your support and feedback mean a great deal to us.
>
> Best regards,
>
> The Authors

---

### Official Review · Reviewer_9E2P · 2024-10-30

**Soundness:** 3
**Presentation:** 3
**Contribution:** 2
**Rating:** 6
**Confidence:** 4

**Summary:**

This paper addresses the inefficiency of deeper layers in Large Language Models (LLMs) by introducing Mix-LN, a novel normalization technique that combines Pre-Layer Normalization (Pre-LN) and Post-Layer Normalization (Post-LN). The authors argue that the widespread use of Pre-LN in models like GPT and LLaMA leads to diminished gradient norms in deeper layers, reducing their effectiveness. Mix-LN applies Post-LN to earlier layers and Pre-LN to deeper layers, ensuring more uniform gradient norms across all layers. Through extensive experiments across various model sizes, the authors demonstrate that Mix-LN consistently outperforms both Pre-LN and Post-LN, promoting more balanced gradient norms throughout the network and enhancing the quality of LLM pre-training and fine-tuning. This approach aims to unlock the full potential of deeper layers in LLMs, improving model capacity without increasing model size.

**Strengths:**

1. This article conducts a good experimental analysis and points out that the gradient norm of Pre-LN and Post-LN shows different trends as the number of layers increases.
2. Comprehensive experiments on both open-weight and in-house models, with multiple evaluation metrics (Angular Distance, Performance Drop, Gradient Norm).
3. Well-structured presentation and logical flow with clear diagrams and mathematical formulations.

**Weaknesses:**

1. **Limited Theoretical Foundation:** The paper lacks rigorous mathematical analysis of Mix-LN's properties, particularly in contrast to well-established theoretical frameworks for methods like DeepNorm[1]. This absence of formal proofs for convergence properties and optimal ratio selection limits our understanding of the method's fundamental principles.

2. **Scale Limitations:** By restricting experiments to models up to 1B parameters and primarily using LLaMA-based architectures, the work lacks demonstrating scalability to production-scale models (7B-175B parameters) or generalizability across different architecture families.

3. **Incremental Innovation:** The proposed Mix-LN solution, while practical, represents a relatively straightforward combination of existing techniques rather than a fundamental advancement in normalization methodology, especially given extensive prior work referenced in related work and other forums[2,3] in this domain.

References:

[1] Wang, H., et al. (2024). "Deepnet: Scaling transformers to 1,000 layers." IEEE TPAMI

[2] Su, J. (2022). "Understanding Pre-Norm vs Post-Norm in Transformers." Scientific Spaces

[3] Raschka, S. (2023). "Layer Normalization Variants in Transformer Architectures." Sebastian Raschka's ML Magazine

**Questions:**

1. **Scaling Properties:** How does Mix-LN perform in very large models (>7B parameters), and what theoretical guarantees can be provided for its effectiveness at scale?

2. **Boundary Dynamics:** What are the theoretical considerations and potential instabilities at the transition point between Pre-LN and Post-LN layers?

3. **Asymptotic Performance:** While accelerated convergence is demonstrated, how does Mix-LN's final performance compare to Post-LN when both are trained to complete convergence under same hyperparameter settings? A controlled study with matched configurations training until performance plateaus would help distinguish whether Mix-LN's benefits extend beyond computational efficiency to fundamental improvements in model capability.

---

> ### Author Response · Authors · 2024-11-20
>
> ### **Response to Reviewer 9E2P [1/4]**
> We would first like to thank you for your time to review our work. We appreciate that you find our experimental analysis good, presentation well-structured and our logical flow clear. We address your comments below.
>
> **W1: Limited Theoretical Foundation: The paper lacks rigorous mathematical analysis of Mix-LN's properties, particularly in contrast to well-established theoretical frameworks for methods like DeepNorm. This absence of formal proofs for convergence properties and optimal ratio selection limits our understanding of the method's fundamental principles.**
>
> - We are fully aware of and deeply respect the foundational theoretical analyses of layer normalization provided by previous works [1,2,3,4]. While the limited rebuttal time window prevents us from presenting a rigorous theoretical framework, we provide the following theoretical analysis aiming to offer insights into the effectiveness of Mix-LN. Note that the theoretical derivation is primarily based on previous works, particularly [1,4].
>
>
> - Denote $x$ as the input of the $l$-th Transformer layer with dimension of $d$. Post-LN applies $\mathrm{LN}(\cdot)$ after the residual addition:
>
>     $$
>     \text{Post-LN}(x) = \mathrm{LN}(x + \mathcal{F}(x)). \quad (1)
>     $$
>
>     In contrast, Pre-LN applies $\mathrm{LN}(\cdot)$ before the residual addition:
>
>     $$
>     \text{Pre-LN}(x) = x + \mathcal{F}(\mathrm{LN}(x)). \quad (2)
>     $$
>
>     As shown in our submission, the derivative of Post-LN and Pre-LN are given:
>
>     $$
>     \frac{\partial \text{Post-LN}(x)}{\partial x} = \frac{\partial \mathrm{LN}(x + \mathcal{F}(x))}{\partial (x + \mathcal{F}(x))}\left(I + \frac{\partial \mathcal{F}(x)}{\partial x} \right), \quad (3)
>     $$
>
>     $$
>     \frac{\partial \text{Pre-LN}(x)}{\partial x} = I + \frac{\partial \mathcal{F}(\mathrm{LN}(x))}{\partial \mathrm{LN}(x)}\frac{\partial \mathrm{LN}(x)}{\partial x}, \quad (4)
>     $$
>
>   Both the above equations involve the Jacobian matrix of layer normalization, $\mathbf{J}_{LN}(x') = \frac{\partial \text{LN}(x')}{\partial x'}$,
>
>
>     where $x'$ is the input of $\mathrm{LN}(\cdot)$. Following Xiong et al. 2020 [1]'s proof, $\mathbf{J}_{LN}(x')$ can be obtained:
>
>
>     $$
>     \frac{\partial \mathrm{LN}(x')}{\partial x'}
>     = \frac{\sqrt{d}}{\| x' \|_2}\bigg(I - \frac{x'x'^\top}{\| x' \|_2^2} \bigg). \quad (5)
>     $$
>
>     Assuming that $x'$ follows a normal distribution with a mean of 0, we have, $\lVert x' \rVert_2 = \sigma_{x'} \sqrt{d}$,
>
>
>     where $\sigma_{x'}$ is the standard deviation of $x'$. Hence,
>
>     $$
>     \frac{\partial \mathrm{LN}(x')}{\partial x'} = \frac{\sqrt{d}}{\sigma_{x'}\sqrt{d}}\bigg(I - \frac{x' x'^\top}{\sigma_{x'}^2 d} \bigg)
>     = \frac{1}{\sigma_{x'}}\bigg(I - \frac{zz^\top}{d} \bigg), \quad (6)
>     $$
>
>     where $z=(x'-\mu_{x'})/\sigma_{x'}$ is the standard normal distribution obtained after layer normalization. Since $d \gg 1$ in LLMs, we can finally obtain:
>
>     $$
>     \frac{\partial \mathrm{LN}(x')}{\partial x'} =  \frac{1}{\sigma_{x'}}I. \quad (7)
>     $$
>
>     In practice, we observe that $\sigma_{x'}$ of Post-LN gradually grows larger than one during training, which means **the spectral norm of the Jacobian matrix of LN is smaller than 1**, i.e.,
>
>     $$
>     \bigg\|\frac{\partial \mathrm{LN}(x')}{\partial x'}\bigg\| = \frac{1}{\sigma_{x'}}  < 1.
>     $$
>
>     According to the derivative of Post-LN in Eq. (3), this down-scaling factor will accumulate as $\prod_{l=1}^{L}\frac{1}{\sigma_{x'}^l}$ over multiple layers $L$, leading to gradient vanishing in early layers in Post-LN transformers.

---

> ### Author Response · Authors · 2024-11-20
>
> ### **Response to Reviewer 9E2P [2/4]**
>
> - **In contrast**, for Pre-LN, Eq. (4) shows that the derivative of the residual connection is decoupled from the term associated with the derivative of $\mathrm{LN}(\cdot)$, preventing vanishing gradients in the early layers. However, because Pre-LN does not normalize the residual connection, the variance of the input to $\mathrm{LN}(\cdot)$, $\sigma_{x'}$, continues to accumulate as the layer depth increases. As a result, Eq. (7) in the deeper layers of Pre-LN approaches zero, causing the right-hand term in Eq. (4) to zero, and leading the derivative of Pre-LN in Eq. (4) to approximate an identity matrix $I$, i.e.,
> $$
> \frac{\partial \text{Pre-LN}(x)}{\partial x} \approx I \quad (8)
> $$
> This indicates that the entire Pre-LN operation in deeper layers fails to contribute effectively during backpropagation. Since the Pre-LN operation encompasses the main components of transformer layers—namely, the attention layer and the feed-forward neural network (FNN) layer—this explains why the deeper layers of Pre-LN tend to contribute less to learning compared to the earlier layers.
>
>
> - Building on the above theoretical analysis, we propose Mix-LN, which replaces the first $\lfloor \alpha L \rfloor$ Pre-LN layers with Post-LN layers. This approach serves two purposes: First, it reduces the number of stacked Pre-LN layers, mitigating the tendency for the derivative of Pre-LN to approach an identity matrix in deeper layers. Second, since Post-LN is applied to only a few layers, the depth is insufficient for the down-scaling factor in Eq. (7) to accumulate to a level where it causes significant gradient vanishing.
>
>
> - The above theoretical analysis aligns perfectly with our observations in Figure 5. We hope that this analysis offers deeper insights into the effectiveness of Mix-LN.
>
>
>     \[1\] Xiong, R., Yang, Y., He, D., Zheng, K., Zheng, S., Xing, C., Zhang, H., Lan, Y., Wang, L. and Liu, T., 2020, November. On layer normalization in the transformer architecture. In *International Conference on Machine Learning* (pp. 10524-10533). PMLR.
>
>     \[2\] Wang, H., Ma, S., Dong, L., Huang, S., Zhang, D. and Wei, F., 2024\. Deepnet: Scaling transformers to 1,000 layers. *IEEE Transactions on Pattern Analysis and Machine Intelligence*.
>
>     \[3\] Liu, Liyuan, Xiaodong Liu, Jianfeng Gao, Weizhu Chen, and Jiawei Han. "Understanding the difficulty of training transformers." *arXiv preprint arXiv:2004.08249* (2020).
>
>     \[4\] Takase, Sho, Shun Kiyono, Sosuke Kobayashi, and Jun Suzuki. "Spike No More: Stabilizing the Pre-training of Large Language Models." *arXiv preprint arXiv:2312.16903* (2023).
>
> **W2: Scale Limitations: By restricting experiments to models up to 1B parameters and primarily using LLaMA-based architectures, the work lacks demonstrating scalability to production-scale models (7B-175B parameters) or generalizability across different architecture families.**
>
> - Thank you for your insightful feedback. We appreciate your suggestion to evaluate our approach on ultra-large models, such as those ranging from 7B to 175B parameters. As you requested, we conduct experiments of LLaMa-7B. All training configurations are identical except for the choice of layer normalization. Due to the limited rebuttal time window, we only finished 13000 steps of training. The perplexity of Mix-LN and Pre-LN is reported in the following table.
>
> - We can clearly see that Mix-LN achieves lower perplexity and faster convergence compared with Pre-LN.
>
>
>     **Table: Perplexity of LLaMA-7B at Different Training Steps**
>
>     | Training Steps | 1000    | 2000 | 5000 | 8000  | 11000 | **13000** |
>     |----------------|---------|------|------|-------|-------|-------|
>     | Pre-LN         | 817.84 | 290.04 | 52.88 | 35.30 | 30.21 | 28.90 |
>     | Mix-LN         | 939.54 | 533.79 | 52.72 | 34.67 | 29.49 | **27.84** |

---

> ### Author Response · Authors · 2024-11-20
>
> ### **Response to Reviewer 9E2P [3/4]**
>
> **W3: Incremental Innovation: The proposed Mix-LN solution, while practical, represents a relatively straightforward combination of existing techniques rather than a fundamental advancement in normalization methodology, especially given extensive prior work referenced in related work and other forums\[2,3\] in this domain.**
>
> - We respectfully disagree with the characterization of our contribution as incremental. While our approach involves a straightforward combination of Pre-LN and Post-LN, the novelty and impact of our work lie far beyond this simplification. Below, we reiterate the distinct contributions of our paper in terms of motivation, insights, and approach.
>
> - **Motivation:** Our motivation is fundamentally novel. While previous research has identified the ineffectiveness of deep layers in large language models (LLMs), these findings have predominantly been leveraged for model compression [1,2,3]. In contrast, we identify this phenomenon as a significant shortcoming of current LLMs, leading to inefficiencies and underutilization of computational resources that could otherwise enhance model performance.
>
> - **Insights:** Building on this novel motivation, our research seeks to uncover the root cause of this inefficiency in deeper layers. This is a non-trivial challenge that has eluded previous studies. For instance, recent work [2] observed that higher layers in current LLMs tend to exhibit high similarity, unlike BERT-style models, which show higher similarity in their shallow layers [4]. However, they incorrectly attributed this behavior to the smaller scale of BERT (e.g., a few hundred million parameters), failing to identify the true underlying cause of this sharp difference. **In contrast to prior work**, we are the first to empirically and theoretically demonstrate that these divergent layer-wise behaviors stem from the choice of layer normalization. Our experiments show that simply altering the layer normalization can replicate these distinct behaviors, even in smaller models with only a few hundred million parameters. Furthermore, our theoretical analysis provides fundamental insights into how Pre-LN and Post-LN differentially influence the training dynamics of earlier and later layers.
>
> - **Approach:** Building on these novel insights, we propose Mix-LN, a simple yet effective layer normalization technique that enhances the functionality of deeper layers, ultimately improving the performance of pre-trained models. While the combination of Pre-LN and Post-LN might appear straightforward, our approach is backed up with both theoretical and empirical analysis. The efficacy of Mix-LN is demonstrated across a wide range of model sizes, from 71M to 7B parameters.
>
> - We humbly argue that papers such as ours, which advance fundamental understanding and provide actionable insights, should not be dismissed due to the simplicity of their methods. On the contrary, we believe the simplicity of our approach, supported by robust evidence, is a strength rather than a limitation.
>
> **Q2: Boundary Dynamics: What are the theoretical considerations and potential instabilities at the transition point between Pre-LN and Post-LN layers?**
>
> - Thank you for bringing up this excellent question! We investigated the transition points between Pre-LN and Post-LN and observed that during training, these layers occasionally experience very large gradients, often referred to as gradient spikes. This phenomenon is understandable given that Post-LN amplifies gradients in the later layers, while Pre-LN amplifies gradients in the earlier layers. Consequently, applying Pre-LN after Post-LN can result in large gradients at the transition layers.
>
> - According to prior studies [5,6], such occasional gradient spikes can negatively impact the final training performance. To address this, we experimented with a straightforward approach: detecting gradients in LN layers whose magnitudes exceed 50 times their current momentum and scaling these gradients down to match their current momentum values. This initial attempt has shown promising results, further improving the training performance of Mix-LN.
>
>     | Method | Mix-LN | Mix-LN w/ Gradient Scaling |
>     | -------- | -------- | --------  |
>     | PPL     | 22.33     | **22.24** |
>
>
> - While we agree that this is a fascinating area worth exploring, a systematic study of gradient spikes introduced by different layer normalization is beyond the scope of our current paper. We believe this topic warrants a dedicated investigation, and we plan to explore it in future work.

---

> ### Author Response · Authors · 2024-11-20
>
> ### **Response to Reviewer 9E2P [4/4]**
>
> **Q3: Asymptotic Performance: While accelerated convergence is demonstrated, how does Mix-LN's final performance compare to Post-LN when both are trained to complete convergence under same hyperparameter settings? A controlled study with matched configurations training until performance plateaus would help distinguish whether Mix-LN's benefits extend beyond computational efficiency to fundamental improvements in model capability.**
>
> - We want to clarify that our evaluation is indeed the final performance of different layer normalization including the full pre-training in Table 1 and supervised fine-tuning in Table 2\. The experiments are exactly controlled experiments with matched configurations with the only difference of layer normalization. Our results show that Mix-LN consistently has fundamental improvements in model capacity over other layer normalization techniques.
>
>     [1] Yin, Lu, You Wu, Zhenyu Zhang, Cheng-Yu Hsieh, Yaqing Wang, Yiling Jia, Gen Li et al. "Outlier weighed layerwise sparsity (owl): A missing secret sauce for pruning llms to high sparsity." *ICML 2024\.*
>
>     [2] Gromov, A., Tirumala, K., Shapourian, H., Glorioso, P. and Roberts, D.A., 2024\. The unreasonable ineffectiveness of the deeper layers. arXiv preprint arXiv:2403.17887.
>
>     [3] Men, X., Xu, M., Zhang, Q., Wang, B., Lin, H., Lu, Y., Han, X. and Chen, W., 2024\. Shortgpt: Layers in large language models are more redundant than you expect. *arXiv preprint arXiv:2403.03853*.
>
>     [4] Sajjad, Hassan, Fahim Dalvi, Nadir Durrani, and Preslav Nakov. "On the effect of dropping layers of pre-trained transformer models." *Computer Speech & Language* 77 (2023): 101429\.
>
>     [5] Chowdhery, A., Narang, S., Devlin, J., Bosma, M., Mishra, G., Roberts, A., Barham, P., Chung, H.W., Sutton, C., Gehrmann, S. and Schuh, P., 2022. Palm: Scaling language modeling with pathways. arXiv 2022. arXiv preprint arXiv:2204.02311, 10.
>
>     [6] Takase, S., Kiyono, S., Kobayashi, S. and Suzuki, J., 2023. Spike No More: Stabilizing the Pre-training of Large Language Models. arXiv preprint arXiv:2312.16903.

---

> ### Author Response · Authors · 2024-11-25
>
> Dear Reviewer 9E2P,
>
> We sincerely thank you for your valuable feedback, which has greatly improved our work. As the end of the discussion phase is approaching, we kindly encourage you to share any additional feedback or updated thoughts you might have regarding our responses and the points raised during the discussion. We highly value your insights, and any further clarification or considerations you provide would greatly contribute to a well-rounded evaluation of our work. Please let us know if there is anything further we can address to help with your assessment.
>
> Kind regards,
>
> The Authors

---

> ### Author Response · Authors · 2024-11-27
> **Kind reminder of our response**
>
> Dear Reviewer 9E2P,
>
> Thank you again for your feedback and your help in improving our work! We'd like to kindly remind you that we've addressed your concerns. As the rebuttal period is still ongoing, we wanted to kindly check if you have any additional questions or concerns that we can address.
>
> For your convenience, we summarize our response here:
>
> - **Theoretical Foundation:** We added theoretical analysis aiming to offer insights into the effectiveness of Mix-LN.
>
> - **Scale Limitations:** We added new experiments with LLaMa-7B.
>
> - **Boundary Dynamics:** We analyzed the potential instability at the transition point and conducted a preliminary experiment to resolve it.
>
> Please let us know if there’s anything further we can address.
>
> Kind regards,
>
> The Authors

---

> ### Comment · Area_Chair_PG2b · 2024-11-30
>
> Dear Reviewer,
>
> Could you kindly respond and indicate whether authors have addressed your concerns?
>
> Thanks, AC

---

> ### Comment · Reviewer_9E2P · 2024-11-30
> **Thanks for your rebuttal. Rating improved.**
>
> Given the thoroughness of both the original submission and rebuttal response, I increase the score for this paper. The work makes good contributions to our understanding of LN in transformer architectures.
>
> Looking forward to the release of the implementation code and experimental procedures, which will further benefit the research community.

---

> > ### Author Response · Authors · 2024-11-30
> > **Thank you once again for your constructive comments and support throughout the review process!**
> >
> > Dear Reviewer 9E2P,
> >
> > Thank you for your thoughtful evaluation of our paper and for raising your score. We deeply appreciate your recognition of the contributions our work makes to the understanding of Layer Normalization in transformer architectures.
> >
> > Best wishes,
> >
> > Authors

---

### Official Review · Reviewer_6By3 · 2024-11-01

**Soundness:** 4
**Presentation:** 3
**Contribution:** 3
**Rating:** 8
**Confidence:** 3

**Summary:**

The paper introduced Mix-LN, which applies Pre-LN and Post-LN in different parts of the LLMs. The author explore the diminished gradient norms of Pre-LN in deeper layers and vanishing gradient of Post-LN in earlier layers, both theoretically and experimentally. They then conduct the experiments comparing Post-LN, DeepNorm, Pre-LN and Mix-LN in pre-training and fine-tuning, with anaylsis on the results. The results show Mix-LN with certain ratios improves the overall capacity and efficiency of LLMs.

**Strengths:**

1. This paper is very well-organized.
2. This paper introduce the problem with theortical anaysis and strengthen with experiments, which makes the problem clear and well-motivated.
3. Comprehensive experiments for presenting the effect of Mix-LN.
4. The anaysis are insightfull.

**Weaknesses:**

1. This figures of angular distance are somewhat non-intuitive. Can't understand on the first sight. Block size more like describing the size of the model than distance between two layers under comparison. The first detailed description is actucally in line 402-403.
2. Typo problem: Figure 3-c (in main body) or Figure 3-e (in caption of Figure 3).

**Questions:**

1. Figure 2-a: Could you explain the triangle-like yellow areas (which also appears but no so significiant in Figure 2-b)? It seems to be discontinuous changes in angular distance between near or adjacent layers. However, from my understanding, the transformer blocks in BERT are the same. Therefore what makes layer 5/6 or 14/15 so special?
2. Figure 2-b: Could you explain the dark-blue line on the right? It seems to be a significiant change of angular distance in last layer.
3. Section 4: Minor comments. There was no introduction for perlexity?
4. Figure 4: a&b shows exact opposite result than Figure 2/3. From my understanding, the angular distance will grow bigger (the color will thus go dark) as the distance between two layers increases. However, for example in layer 10 of Figure 4-b, the farthes has smallest angular distance. Please correct me if I am mistaken.

---

> ### Author Response · Authors · 2024-11-20
>
> ### **Response to Reviewer 6By3**
> We are really grateful for your valuable review! We appreciate it for your positive score and detailed comments. We are glad that you find our paper well-organized and insightful. We address the weakness pointed out by you one by one as follows:
>
> **W1: This figures of angular distance are somewhat non-intuitive. Can't understand on the first sight. Block size more like describing the size of the model than distance between two layers under comparison. The first detailed description is actucally in line 402-403.**
>
> - We appreciate the reviewer bringing up this important point. In our work, we initially followed the terminology of angular distance as used in prior research (https://arxiv.org/pdf/2403.17887), which is why we adopted their phrasing.  However, we fully agree with your concern regarding the potential for misunderstanding caused by the term "Block size." To address this, we have updated the terminology to "subsequent $n^{th}$ layer" in our revision.
>
> **W2: Typo problem: Figure 3-c (in main body) or Figure 3-e (in caption of Figure 3).**
> - Thanks! We have fixed them in our revision.
>
> **Q1: Figure 2-a: Could you explain the triangle-like yellow areas (which also appears but no so significiant in Figure 2-b)? It seems to be discontinuous changes in angular distance between near or adjacent layers. However, from my understanding, the transformer blocks in BERT are the same. Therefore what makes layer 5/6 or 14/15 so special?**
>
> - Great observation! This is indeed a fascinating phenomenon. We believe it is strongly correlated with the findings that different structural information is learned at varying depths of the language model.  For instance, (https://aclanthology.org/P19-1356.pdf) discovered that BERT captures a rich hierarchy of linguistic information: surface-level features are learned in the lower layers, syntactic features in the middle layers, and semantic features in the higher layers. Despite the transformer blocks in BERT having identical architectures, different hierarchical information is encoded across distinct groups of layers. Layers within the same group tend to learn similar features, contributing to the compositional nature of the representations learned by BERT.
>
> - A similar phenomenon has recently been observed in state-of-the-art (SOTA) LLMs such as Pythia, GPT-2, and Phi (https://arxiv.org/pdf/2406.19384), which also aligns with triangle-like areas in Figure 2-b for LLaMA.
>
>
>
> **Q2: Figure 2-b: Could you explain the dark-blue line on the right? It seems to be a significiant change of angular distance in last layer.**
> - The dard-blue line on the right represents the Angular Distance between the last layer with previous layers. It is common that the last layer has very different representations with other layers, as the features from the last layer are task-specific and optimized to directly facilitate the model's objective. The representations from lower and intermediate layers are rich in terms of general contextual information, while the final layer is tasked with deciding or summarizing based on this information to produce an output. As a result, the last layer’s features may lose some of the nuanced intermediate representations and instead focus on solving the specific task, making them less similar to the features from other layers.
>
> **Q3: Section 4: Minor comments. There was no introduction for perlexity?**
> - Perplexity is a fundamental metric used to evaluate the quality of language models. It is commonly used to assess the ability of a language model to predict a sequence of words or tokens, serving as an indicator of how well the model is capturing the statistical properties of natural language. Formally, perplexity can be defined as the exponential of the average negative log-likelihood of a test set:
>     $$
>     \text{Perplexity} = \exp\left( - \frac{1}{N} \sum_{i=1}^{N} \log P(T_i \mid T_{1}, \dots, T_{i-1}) \right)
>     $$
>
>
> **Q4: Figure 4: a&b shows exact opposite result than Figure 2/3. From my understanding, the angular distance will grow bigger (the color will thus go dark) as the distance between two layers increases. However, for example in layer 10 of Figure 4-b, the farthes has smallest angular distance. Please correct me if I am mistaken.**
>
> - The difference between figure 4 and figure 2/3 is that figure 4 depicts the row-normalized angular distance. Following (https://arxiv.org/pdf/2403.17887), normalizing the angular distance by row can tell us the least important stack of layers (the optimal block to prune) for a given block size n (lightest yellow). Figure 4 demonstrates that later layers of Pre-LN (Figure 4-b) has larger yellow areas than Mix-LN (Figure 4-a), indicating that Mix-LN’s deeper layers in general more important than deeper layers of Pre-LN.

---

> > ### Comment · Reviewer_6By3 · 2024-11-29
> >
> > Thank you for your detailed reply. Your answers are helpful and resolve my problems. I will keep my rating.

---

> > > ### Author Response · Authors · 2024-11-29
> > > **Thank you for your recognition and support!**
> > >
> > > Dear Reviewer 6By3,
> > >
> > > We sincerely thank you for recognizing the merits of our paper. I am glad that our response resolves your problems! Your support means a lot to us!
> > >
> > > Best wishes,
> > >
> > > Authors

---

### Official Review · Reviewer_CgHw · 2024-11-03

**Soundness:** 2
**Presentation:** 3
**Contribution:** 2
**Rating:** 5
**Confidence:** 4

**Summary:**

The paper proposes Mix-LN, a hybrid normalization approach that combines Pre-Layer Normalization (Pre-LN) and Post-Layer Normalization (Post-LN) within large language models (LLMs). The technique leverages Post-LN in shallow layers to mitigate gradient vanishing and Pre-LN in deeper layers to maintain gradient flow, with the aim of maximizing layer effectiveness throughout the network. While Mix-LN demonstrates some empirical gains, especially in mid-sized LLMs, the contribution remains incremental in the broader landscape of normalization techniques for LLMs. The paper’s main contributions are primarily empirical, and it lacks a strong theoretical foundation to substantiate why Mix-LN improves gradient dynamics compared to recent normalization methods.

**Strengths:**

- Novel Approach to Gradient Flow in LLMs: Mix-LN’s hybrid approach to layer normalization is unique, attempting to combine the advantages of Pre-LN and Post-LN across different model depths.

- Good Experimental Validation: The paper includes extensive experimentation on multiple model sizes and tasks, showing consistent improvement with Mix-LN, particularly in mid-sized models.

- Improved Training Stability: Mix-LN appears to mitigate some of the training instability issues commonly observed with Post-LN in large models, an advantage for practitioners.

**Weaknesses:**

- Lack of Theoretical Rigor: The paper does not provide a detailed theoretical framework to explain why Mix-LN achieves balanced gradient dynamics across layers, which is crucial for the paper’s validity. In the well-explored field of normalization for LLMs, incremental changes require more substantial theoretical backing to make a notable contribution.

- **Limited Comparison** with State-of-the-Art Normalization Techniques: The paper lacks direct comparison with recent normalization methods, such as Admin or Sandwich LN, which also address deep-layer gradient inefficiencies. Without these comparisons, it’s challenging to assert Mix-LN’s effectiveness over other recent innovations.

- **Diminished Gains on Very Large Models**: The benefits of Mix-LN become less pronounced in very large models, such as LLaMA-1B, where performance gains are smaller. This suggests potential scalability issues for Mix-LN in ultra-large models like 7B, 13B and so on, which is a limitation given the trajectory of LLM research.

- Applicability Limited to LLMs: Mix-LN has only been evaluated on LLMs, and its effectiveness on non-language models or other architectures remains untested. This limits the broader applicability and impact of the approach.

- Hyperparameter Sensitivity Not Thoroughly Explored: The paper does not fully address the sensitivity of the hyperparameter 𝛼 (controlling the transition point between Pre-LN and Post-LN), which could impact Mix-LN’s practical usability across different settings.

**Questions:**

- Could the authors provide more insights or theoretical justifications for why Mix-LN enhances gradient flow differently across shallow and deep layers?

- How does Mix-LN perform when compared to other recent normalization methods, such as Admin or Sandwich LN?

- How sensitive is Mix-LN to the hyperparameter 𝛼, and is this value stable across different tasks and model scales?

- Can the authors clarify Mix-LN’s computational impact during training? Does it introduce any additional training or inference overhead?

---

> ### Author Response · Authors · 2024-11-20
>
> ### **Response to Reviewer CgHw [1/4]**
>
> We thank the reviewer for the time spent on reviewing our submission. We are glad that you found our approach novel. We address the weakness pointed out by you one by one as follows:
>
> **W0： While Mix-LN demonstrates some empirical gains, especially in mid-sized LLMs, the contribution remains incremental in the broader landscape of normalization techniques for LLMs.**
>
> - We respectfully disagree with the characterization of our contribution as incremental. While our approach involves a straightforward combination of Pre-LN and Post-LN, the novelty and impact of our work lie far beyond this simplification. Below, we reiterate the distinct contributions of our paper in terms of motivation, insights, and approach.
>
>  - **Motivation:** Our motivation is fundamentally novel. While previous research has identified the ineffectiveness of deep layers in large language models (LLMs), these findings have predominantly been leveraged for model compression [1,2,3]. In contrast, we identify this phenomenon as a significant shortcoming of current LLMs, leading to inefficiencies and underutilization of computational resources that could otherwise enhance model performance.
>
>  - **Insights:** Building on this novel motivation, our research seeks to uncover the root cause of this inefficiency in deeper layers. This is a non-trivial challenge that has eluded previous studies. For instance, recent work [2] observed that higher layers in current LLMs tend to exhibit high similarity, unlike BERT-style models, which show higher similarity in their shallow layers [4]. However, they incorrectly attributed this behavior to the smaller scale of BERT (e.g., a few hundred million parameters), failing to identify the true underlying cause of this sharp difference. **In contrast to prior work**, we are the first to empirically and theoretically demonstrate that these divergent layer-wise behaviors stem from the choice of layer normalization. Our experiments show that simply altering the layer normalization can replicate these distinct behaviors, even in smaller models with only a few hundred million parameters. Furthermore, our theoretical analysis provides fundamental insights into how Pre-LN and Post-LN differentially influence the training dynamics of earlier and later layers.
>
> - **Approach:** Building on these novel insights, we propose Mix-LN, a simple yet effective layer normalization technique that enhances the functionality of deeper layers, ultimately improving the performance of pre-trained models. While the combination of Pre-LN and Post-LN might appear straightforward, our approach is backed up with both theoretical and empirical analysis. The efficacy of Mix-LN is demonstrated across a wide range of model sizes, from 71M to 7B parameters.
>
> We humbly argue that papers such as ours, which advance fundamental understanding and provide actionable insights, should not be dismissed due to the simplicity of their methods. On the contrary, we believe the simplicity of our approach, supported by robust evidence, is a strength rather than a limitation.

---

> ### Author Response · Authors · 2024-11-20
>
> ### **Response to Reviewer CgHw [2/4]**
> **W1: Provide insights/theoretical justifications for why Mix-LN enhances gradient flow differently across shallow and deep layers?**
>
> - We are fully aware of and deeply respect the foundational theoretical analyses of layer normalization provided by previous works [1,2,3,4]. While the limited rebuttal time window prevents us from presenting a rigorous theoretical framework, we provide the following theoretical analysis aiming to offer insights into the effectiveness of Mix-LN. Note that the theoretical derivation is primarily based on previous works, particularly [1,4].
>
>     Denote $x$ as the input of the $l$-th Transformer layer with dimension of $d$. Post-LN applies $\mathrm{LN}(\cdot)$ after the residual addition:
>
>     $$
>     \text{Post-LN}(x) = \mathrm{LN}(x + \mathcal{F}(x)). \quad (1)
>     $$
>
>     In contrast, Pre-LN applies $\mathrm{LN}(\cdot)$ before the residual addition:
>
>     $$
>     \text{Pre-LN}(x) = x + \mathcal{F}(\mathrm{LN}(x)). \quad (2)
>     $$
>
>     As shown in our submission, the derivative of Post-LN and Pre-LN are given:
>
>     $$
>     \frac{\partial \text{Post-LN}(x)}{\partial x} = \frac{\partial \mathrm{LN}(x + \mathcal{F}(x))}{\partial (x + \mathcal{F}(x))}\left(I + \frac{\partial \mathcal{F}(x)}{\partial x} \right), \quad (3)
>     $$
>
>     $$
>     \frac{\partial \text{Pre-LN}(x)}{\partial x} = I + \frac{\partial \mathcal{F}(\mathrm{LN}(x))}{\partial \mathrm{LN}(x)}\frac{\partial \mathrm{LN}(x)}{\partial x}, \quad (4)
>     $$
>
>     Both the above equations involve the Jacobian matrix of layer normalization, $\mathbf{J}_{LN}(x') = \frac{\partial \text{LN}(x')}{\partial x'}$,
>
>
>     where $x'$ is the input of $\mathrm{LN}(\cdot)$. Following Xiong et al. 2020 [1]'s proof, $\mathbf{J}_{LN}(x')$ can be obtained:
>
>
>     $$
>     \frac{\partial \mathrm{LN}(x')}{\partial x'}
>     = \frac{\sqrt{d}}{\| x' \|_2}\bigg(I - \frac{x'x'^\top}{\| x' \|_2^2} \bigg). \quad (5)
>     $$
>
>     Assuming that $x'$ follows a normal distribution with a mean of 0, we have, $\lVert x' \rVert_2 = \sigma_{x'} \sqrt{d}$,
>
>
>     where $\sigma_{x'}$ is the standard deviation of $x'$. Hence,
>
>     $$
>     \frac{\partial \mathrm{LN}(x')}{\partial x'} = \frac{\sqrt{d}}{\sigma_{x'}\sqrt{d}}\bigg(I - \frac{x' x'^\top}{\sigma_{x'}^2 d} \bigg)
>     = \frac{1}{\sigma_{x'}}\bigg(I - \frac{zz^\top}{d} \bigg), \quad (6)
>     $$
>
>     where $z=(x'-\mu_{x'})/\sigma_{x'}$ is the standard normal distribution obtained after layer normalization. Since $d \gg 1$ in LLMs, we can finally obtain:
>
>     $$
>     \frac{\partial \mathrm{LN}(x')}{\partial x'} =  \frac{1}{\sigma_{x'}}I. \quad (7)
>     $$
>
>     In practice, we observe that $\sigma_{x'}$ of Post-LN gradually grows larger than one during training, which means **the spectral norm of the Jacobian matrix of LN is smaller than 1**, i.e.,
>
>     $$
>     \bigg\|\frac{\partial \mathrm{LN}(x')}{\partial x'}\bigg\| = \frac{1}{\sigma_{x'}}  < 1.
>     $$
>     According to the derivative of Post-LN in Eq. (3), this down-scaling factor will accumulate as $\prod_{l=1}^{L}\frac{1}{\sigma_{x'}^l}$ over multiple layers $L$, leading to gradient vanishing in early layers in Post-LN transformers.
>
> - **In contrast**, for Pre-LN, Eq. (4) shows that the derivative of the residual connection is decoupled from the term associated with the derivative of $\mathrm{LN}(\cdot)$, preventing vanishing gradients in the early layers. However, because Pre-LN does not normalize the residual connection, the variance of the input to $\mathrm{LN}(\cdot)$, $\sigma_{x'}$, continues to accumulate as the layer depth increases. As a result, Eq. (7) in the deeper layers of Pre-LN approaches zero, causing the right-hand term in Eq. (4) to zero, and leading the derivative of Pre-LN in Eq. (4) to approximate an identity matrix $I$, i.e.,
> $$
> \frac{\partial \text{Pre-LN}(x)}{\partial x} \approx I \quad   (8)
> $$
> This indicates that the entire Pre-LN operation in deeper layers fails to contribute effectively during backpropagation. Since the Pre-LN operation encompasses the main components of transformer layers—namely, the attention layer and the feed-forward neural network (FNN) layer—this explains why the deeper layers of Pre-LN tend to contribute less to learning compared to the earlier layers.
> - Building on the above theoretical analysis, we propose Mix-LN, which replaces the first $\lfloor \alpha L \rfloor$ Pre-LN layers with Post-LN layers. This approach serves two purposes: First, it reduces the number of stacked Pre-LN layers, mitigating the tendency for the derivative of Pre-LN to approach an identity matrix in deeper layers. Second, since Post-LN is applied to only a few layers, the depth is insufficient for the down-scaling factor in Eq. (7) to accumulate to a level where it causes significant gradient vanishing.
> - The above theoretical analysis aligns perfectly with our observations in Figure 5. We hope that this analysis offers deeper insights into the effectiveness of Mix-LN

---

> ### Author Response · Authors · 2024-11-20
>
> ### **Response to Reviewer CgHw [3/4]**
>
> **W2： Limited Comparison with State-of-the-Art Normalization Techniques: The paper lacks direct comparison with recent normalization methods, such as Admin or Sandwich LN, which also address deep-layer gradient inefficiencies. Without these comparisons, it’s challenging to assert Mix-LN’s effectiveness over other recent innovations.**
>
> - As requested, we conducted comparisons using LLaMA-250M to evaluate Mix-LN against recent normalization methods, including Admin [5], Sandwich-LN [6], and Group-LN [7,8]. The results indicate that Sandwich-LN and Group-LN slightly outperform Pre-LN, while Admin performs worse. However, all of these methods fall to reduce perplexity below 23, falling short of Mix-LN. This result highlights the effectiveness of Mix-LN compared to other recent innovations.
>
>     | Model         | Pre-LN | Admin  | Group-LN | Sandwich-LN | Mix-LN  |
>     | :------------ | :----- | :----- | :------- | :---------- | :------ |
>     | LLaMA-250M    | 23.39  | 24.82  | 23.10    | 23.26       | **22.33** |
>
> **W3: Diminished Gains on Very Large Models: The benefits of Mix-LN become less pronounced in very large models, such as LLaMA-1B, where performance gains are smaller. This suggests potential scalability issues for Mix-LN in ultra-large models like 7B, 13B and so on, which is a limitation given the trajectory of LLM research.**
>
> - First, we would like to clarify that the small performance gain reported in our submission of 1B model is due to a suboptimal choice of α. Specifically, for the 1B model, the ratio α was set as 0.33, rather than the optimal ratio of 0.25 as stated in our claims. When α is set to the optimal value of 0.25, Mix-LN achieves a notable performance improvement over Pre-LN, with a nearly 0.5 perplexity reduction for LLaMa-1B as demonstrated in the following table. Note that it is reasonable for the performance gains to reduced to certain extends as the model size increases, since larger models inherently have better baseline performance, reducing the room for further improvement.
>
>
>     | LLaMa-1B | Pre-LN | Mix-LN (α=0.33) | Mix-LN (α=0.25) |
>     | :---- | :---- | :---- | :---- |
>     | PPL | 18.65 | 18.40 | 18.18 |
>
> - More importantly, the improvement in perplexity translates into significant enhancements in downstream tasks, including self-supervised fine-tuning (SFT) and reinforcement learning from human feedback (RLHF). For instance, after applying SFT with the trained 1B models on the Commonsense170K dataset, Mix-LN achieves an average accuracy improvement of 1.24%, with a particularly remarkable gain of 5.42% on the Winogrande task.
>
>     | Method | MMLU | BoolQ | ARC-e | PIQA | Hellaswag | OBQA | Winogrande | Avg. |
>     | :---- | :---- | :---- | :---- | :---- | :---- | :---- | :---- | :---- |
>     | Pre-LN | 26.54 | 62.20 | 45.70 | 67.79 | 30.96 | 17.40 | 50.51 | 43.01 |
>     | Mix-LN | 27.99 | 61.93 | 48.11 | 68.50 | 31.35 | 18.80 | 55.93 | 44.25 |
>
> - Consistently, the benefits of Mix-LN can be seamlessly transferred to RLHF. Following Adam-mini (https://arxiv.org/pdf/2406.16793), we use the ultrafeedback dataset and implement the RLHF workflow from InstructGPT (https://arxiv.org/abs/2203.02155) to optimize the preference reward. Mix-LN achieves a notable reward gain (higher is better) over Pre-LN, i.e., 1.32 vs 0.75.
>
>     | Method | Model | Final Reward |
>     | :---- | :---- |:---- |
>     | Pre-LN |  LLaMA-1B | 0.75 |
>     | Mix-LN | LLaMA-1B | 1.32 |
>
> - To further address your concerns regarding the benefits of Mix-LN on large-scale models, we conducted experiments with **LLaMa-7B** using the same setup as GaLore. All training configurations were kept identical, except for the choice of layer normalization. Given the limited rebuttal time window and our computing resources, we only completed 13,000 steps of training. However, based on our experience, models that exhibit consistent improvements early in training typically retain these advantages through the later stages. The perplexity of Mix-LN and Pre-LN under these conditions is summarized in the table below. We also report the training curves of Mix-LN and Pre-LN in Figure 4 of our revision.
>
>     **Table: Perplexity of LLaMA-7B at Different Training Steps**
>
>     | Training Steps | 1000    | 2000 | 5000 | 8000  | 11000 | **13000** |
>     |----------------|---------|------|------|-------|-------|-------|
>     | Pre-LN         | 817.84 | 290.04 | 52.88 | 35.30 | 30.21 | 28.90 |
>     | Mix-LN         | 939.54 | 533.79 | 52.72 | 34.67 | 29.49 | **27.84** |

---

> > ### Comment · Reviewer_CgHw · 2024-11-25
> >
> > Thank you for providing evluation resulsts in LLaMA-7B. And also resolve some of my other questions. I will raise my ratings.

---

> ### Author Response · Authors · 2024-11-20
>
> ### **Response to Reviewer CgHw [4/4]**
> **W4: Applicability Limited to LLMs: Mix-LN has only been evaluated on LLMs, and its effectiveness on non-language models or other architectures remains untested. This limits the broader applicability and impact of the approach.**
>
> - To evaluate Mix-LN on non-language tasks, we replaced Pre-LN in ViT models with Mix-LN with α=0.25 and trained the updated model on ImageNet-1K, following the ConvNeXt training configurations for 120 epochs. The results clearly demonstrate that the benefits of Mix-LN also generalize to vision tasks. Notably, the performance gains are more pronounced for larger model (ViT-Small) than smaller model (ViT-Tiny).
>
>     |  | Pre-LN | Mix-LN |
>     | :---- | :---- | :---- |
>     | ViT-Tiny | 67.30 | **67.34** |
>     | ViT-Small | 75.99 | **76.40** |
>
> **W5: How sensitive is Mix-LN to the hyperparameter 𝛼, and is this value stable across different tasks and model scales?**
>
> - Mix-LN is quite robust to the choice of 𝛼 as we have reported in Table 3 of our submission with LLaMA-250M. To draw a more solid conclusion, we added an extra experiment with the LLaMa-1B model. The results are summarized in the table below, confirming that Mix-LN remains robust to the choice of α. Results that outperform Pre-LN are highlighted in bold for clarity. Specifically, as long as α is constrained below 0.5, Mix-LN consistently outperforms Pre-LN, with the best results achieved at α=0.25.
>
>     **Table: Perplexity of LLaMA-250M with various Post-LN ratios α**
>     | Post-LN Ratios (α) | Pre-LN | Mix-LN (12.5%) | Mix-LN (25.0%) | Mix-LN (33.3%) | Mix-LN (41.7%) | Mix-LN (50.0%) | Mix-LN (75.0%) | Post-LN |
>     |---------------------|--------|----------------|----------------|----------------|----------------|----------------|----------------|---------|
>     | **Perplexity**      | 23.39  | **22.37**          | **22.33**      | **22.83**          | **22.80**          | **22.81**          | 23.64          | 32.18   |
>
>
>     **Table: Perplexity of LLaMA-1B with various Post-LN ratios α**
>     | Post-LN Ratios (α)  | Pre-LN | Mix-LN (16.7%) | Mix-LN (25.0%) | Mix-LN (33.3%) | Mix-LN (41.7%) | Mix-LN (50.0%) | Post-LN |
>     |----------------|--------|--------|--------|--------|---------|---------|---------|
>     | **Perplexity**        | 18.65 | **18.34**  | **18.18**  | **18.41**  | **18.55**   | 18.86   | 1434    |
>
>
> **Q6: Can the authors clarify Mix-LN’s computational impact during training? Does it introduce any additional training or inference overhead?**
>
> - Thanks for pointing out this great question. Mix-LN does not introduce any additional training and inference overhead compared to pure Post-LN and Pre-LN. This is one of the advantages of Mix-LN over previous LN variants such as Admin and Sandwich LN.
>
> - The table below shows the time per iteration for LLaMa-250M on the A800. It can be observed that, compared to pre-layer normalization and post-layer normalization, the mix approach does not introduce any additional overhead. However, the Sandwich LN incurs additional overhead due to the increased number of uses of layer normalization.
>
>
>     |  Pre-LN |  Post-LN | Mix-LN | Sandwich-LN |
>     | -------- | -------- | -------- |-------- |
>     | 1.97s/iter     | 1.98s/iter     |   1.97s/iter   | 2.42s/iter|
>
>
>
>     \[1\] Xiong, R., Yang, Y., He, D., Zheng, K., Zheng, S., Xing, C., Zhang, H., Lan, Y., Wang, L. and Liu, T., 2020, November. On layer normalization in the transformer architecture. In *International Conference on Machine Learning* (pp. 10524-10533). PMLR.
>
>     \[2\] Wang, H., Ma, S., Dong, L., Huang, S., Zhang, D. and Wei, F., 2024\. Deepnet: Scaling transformers to 1,000 layers. *IEEE Transactions on Pattern Analysis and Machine Intelligence*.
>
>     \[3\] Liu, Liyuan, Xiaodong Liu, Jianfeng Gao, Weizhu Chen, and Jiawei Han. "Understanding the difficulty of training transformers." *arXiv preprint arXiv:2004.08249* (2020).
>
>     \[4\] Takase, Sho, Shun Kiyono, Sosuke Kobayashi, and Jun Suzuki. "Spike No More: Stabilizing the Pre-training of Large Language Models." *arXiv preprint arXiv:2312.16903* (2023).
>
>     [5] Liu, L., Liu, X., Gao, J., Chen, W. and Han, J., 2020. Understanding the difficulty of training transformers. arXiv preprint arXiv:2004.08249.
>
>     [6] Ding, M., Yang, Z., Hong, W., Zheng, W., Zhou, C., Yin, D., Lin, J., Zou, X., Shao, Z., Yang, H. and Tang, J., 2021. Cogview: Mastering text-to-image generation via transformers. Advances in neural information processing systems, 34, pp.19822-19835.
>
>     [7] Wu, Y. and He, K., 2018. Group normalization. In Proceedings of the European conference on computer vision (ECCV) (pp. 3-19).
>
>     [8] Ma, X., Yang, X., Xiong, W., Chen, B., Yu, L., Zhang, H., May, J., Zettlemoyer, L., Levy, O. and Zhou, C., 2024. Megalodon: Efficient llm pretraining and inference with unlimited context length. arXiv preprint arXiv:2404.08801.

---

> ### Author Response · Authors · 2024-11-27
> **We sincerely appreciate your response and the increase in your score!**
>
> Dear Reviewer CgHw,
>
> Thank you for your kind response and for raising your rating. We sincerely appreciate your recognition of our efforts in addressing your questions and providing the evaluation results for LLaMA-7B.
>
> We noticed that the revised score remains at a borderline level, and we would like to kindly ask if you have any remaining concerns or unresolved questions about our work. If there are specific points you feel need further clarification or improvement, we would be more than happy to address them in detail.
>
> Your feedback is invaluable to us, and we are committed to ensuring our paper meets the highest standards. Thank you again for your time, thoughtful review, and support.
>
> Best regards,
>
> The Authors

---

### Official Review · Reviewer_iHrq · 2024-11-03

**Soundness:** 2
**Presentation:** 3
**Contribution:** 2
**Rating:** 6
**Confidence:** 4

**Summary:**

In this paper the authors context the employment of either pre-layer normalization (Pre-LN) and post layer normalization. (Post-LN) They bring an argument showing vanishing gradient for earlier or later layers. Building on this intuition, they propose to place post-LN in early layers and pre-LN in later ones, to maximize the gradient. The evaluation is performed on LLMs, either large and small scale ones.

**Strengths:**

- The observation related to gradient norm brought by LN is interesting, clear, and straight to the point.
- Part of the quantitative evaluation is conducted to large-scale LLMs, making this work actual.
- All the empirical evaluation is quite realistic and likely correct.
- Figures in general do an excellent job in providing a display on the distributions.

**Weaknesses:**

- The relative gains in terms of perplexity/accuracy for larger model slims down. This in a certain sense contradicts the vanishing gradient argument. Besides, the gap compared to pre-LM becomes slower and slower.
- The success of the approach depends on $\alpha$: there are cases like Llama-1B for BoolQ in which only pre-LM performs better, indicating that tuning $\alpha$ properly can be determinant.
- Metrics are not always complete: I could not find, for example, performance drop when using Mix-LN

**Questions:**

- How does the performance drop when using Mix-LN?
- Is this approach still valid for other tasks (like for example image classification, with a Transformer architecture on ImageNet-1k)?
- Have the authors attempted to compare with other normalization techniques (like group normalization)?
- Can the authors provide theoretical boundaries to the vanishing/exploding gradient conditions for equations (3) and (4)?

---

> ### Author Response · Authors · 2024-11-20
>
> ### **Response to Reviewer iHrq [1/3]**
>
> We would first like to thank you for your time and effort in reviewing our work. We are glad that you have found our observation interesting, clear, and straight to the point, and our evaluation is realistic and likely correct. We would like to address the weakness pointed out by you one by one as follows:
>
> **W1: The relative gains in terms of perplexity/accuracy for larger model slims down. This in a certain sense contradicts the vanishing gradient argument. Besides, the gap compared to pre-LM becomes slower and slower.**
>
> - First, we would like to clarify that the small performance gain reported in our submission of 1B model is due to a suboptimal choice of α. Specifically, for the 1B model, the ratio α was set as 0.33, rather than the optimal ratio of 0.25 as stated in our claims. When α is set to the optimal value of 0.25, Mix-LN achieves a notable performance improvement over Pre-LN, with a nearly 0.5 perplexity reduction for LLaMa-1B as demonstrated in the following table. Note that it is reasonable for the performance gains to reduced to certain extends as the model size increases, since larger models inherently have better baseline performance, reducing the room for further improvement.
>
>     | LLaMa-1B | Pre-LN | Mix-LN (α=0.33) | Mix-LN (α=0.25) |
>     | :---- | :---- | :---- | :---- |
>     | PPL | 18.65 | 18.40 | 18.18 |
>
> - More importantly, the improvement in perplexity translates into significant enhancements in downstream tasks, including self-supervised fine-tuning (SFT) and reinforcement learning from human feedback (RLHF). For instance, after applying SFT with the trained 1B models on the Commonsense170K dataset, Mix-LN achieves an average accuracy improvement of 1.24%, with a particularly remarkable gain of 5.42% on the Winogrande task.
>
>     | Method | MMLU | BoolQ | ARC-e | PIQA | Hellaswag | OBQA | Winogrande | Avg. |
>     | :---- | :---- | :---- | :---- | :---- | :---- | :---- | :---- | :---- |
>     | Pre-LN | 26.54 | 62.20 | 45.70 | 67.79 | 30.96 | 17.40 | 50.51 | 43.01 |
>     | Mix-LN | 27.99 | 61.93 | 48.11 | 68.50 | 31.35 | 18.80 | 55.93 | 44.25 |
>
> - Consistently, the benefits of Mix-LN can be seamlessly transferred to RLHF. Following Adam-mini (https://arxiv.org/pdf/2406.16793), we use the ultrafeedback dataset and implement the RLHF workflow from InstructGPT (https://arxiv.org/abs/2203.02155) to optimize the preference reward. Mix-LN achieves a notable reward gain (higher is better) over Pre-LN, i.e., 1.32 vs 0.75.
>
>     | Method | Model | Final Reward |
>     | :---- | :---- |:---- |
>     | Pre-LN |  LLaMA-1B | 0.75 |
>     | Mix-LN | LLaMA-1B | 1.32 |
>
> - To further address your concerns regarding the benefits of Mix-LN on large-scale models, we conducted experiments with **LLaMa-7B** using the same setup as GaLore. All training configurations were kept identical, except for the choice of layer normalization. Given the limited rebuttal time window and our computing resources, we only completed 13,000 steps of training. However, based on our experience, models that exhibit consistent improvements early in training typically retain these advantages through the later stages. The perplexity of Mix-LN and Pre-LN under these conditions is summarized in the table below. We also report the training curves of Mix-LN and Pre-LN in Figure 4 of our revision.
>
>     **Table: Perplexity of LLaMA-7B at Different Training Steps**
>
>     | Training Steps | 1000    | 2000 | 5000 | 8000  | 11000 | **13000** |
>     |----------------|---------|------|------|-------|-------|-------|
>     | Pre-LN         | 817.84 | 290.04 | 52.88 | 35.30 | 30.21 | 28.90 |
>     | Mix-LN         | 939.54 | 533.79 | 52.72 | 34.67 | 29.49 | **27.84** |

---

> ### Author Response · Authors · 2024-11-20
>
> ### **Response to Reviewer iHrq [2/3]**
>
>
> **W2: The success of the approach depends on α: there are cases like Llama-1B for BoolQ in which only pre-LM performs better, indicating that tuning α properly can be determinant**
>
> - While Mix-LN slightly underperforms Pre-LN on the BoolQ dataset, it consistently outperforms Pre-LN on the remaining 6 datasets, as shown in the table above. Summarizing the results from our paper, Mix-LN achieves better performance than Pre-LN in **17** out of **18** evaluations, with BoolQ being the sole exception. Therefore, we believe it is reasonable to conclude that Mix-LN does not result in performance degradation but instead serves as a performance booster for LLMs.
>
> - Additionally, we want to emphasize that Mix-LN demonstrates strong robustness to the choice of α, as shown in Table 3 of our submission with LLaMA-250M. To further substantiate this, we conducted an additional experiment with the LLaMA-1B model. The results are summarized in the table below, confirming that Mix-LN remains robust to the choice of α. Results that outperform Pre-LN are highlighted in bold for clarity. Specifically, as long as α is constrained below 0.5, Mix-LN consistently outperforms Pre-LN, with the best results achieved at α=0.25.
>
>     **Table: Perplexity of LLaMA-250M with various Post-LN ratios α**
>     | Post-LN Ratios (α) | Pre-LN | Mix-LN (12.5%) | Mix-LN (25.0%) | Mix-LN (33.3%) | Mix-LN (41.7%) | Mix-LN (50.0%) | Mix-LN (75.0%) | Post-LN |
>     |---------------------|--------|----------------|----------------|----------------|----------------|----------------|----------------|---------|
>     | **Perplexity**      | 23.39  | **22.37**          | **22.33**      | **22.83**          | **22.80**          | **22.81**          | 23.64          | 32.18   |
>
>     ---
>
>     **Table: Perplexity of LLaMA-1B with various Post-LN ratios α**
>     | Post-LN Ratios (α)  | Pre-LN | Mix-LN (16.7%) | Mix-LN (25.0%) | Mix-LN (33.3%) | Mix-LN (41.7%) | Mix-LN (50.0%) | Post-LN |
>     |----------------|--------|--------|--------|--------|---------|---------|---------|
>     | **Perplexity**        | 18.65 | **18.34**  | **18.18**  | **18.41**  | **18.55**   | 18.86   | 1434    |
>
> **W3: Metrics are not always complete: I could not find, for example, performance drop when using Mix-LN. Q1：How does the performance drop when using Mix-LN?**
>
> - Thank you for bringing up this issue. We have added the performance drop of the 130M model on the ARC-e task, and the results are presented in the table below. The data shows that the performance drop for Mix-LN and Pre-LN follows similar trends overall, as most layers in Mix-LN are still based on Pre-LN. However, removing deeper layers from Mix-LN consistently results in larger performance drops compared to Pre-LN, indicating that the later layers contribute more significantly to the model's performance.
>
>
>     | Method   | Layer 1  | Layer 2  | Layer 3  | Layer 4  | Layer 5  | Layer 6  | Layer 7  | Layer 8  | Layer 9  | Layer 10 | Layer 11 | Layer 12 |
>     |---------|----------|----------|----------|----------|----------|----------|----------|----------|----------|----------|----------|----------|
>     | Mix-LN     | -14.56   | -14.54   | -1.94    | -5.43    | -0.29    | -0.38     | -1.05    | -1.09    | -2.72    | -0.75    | -2.06    | -3.07    |
>     | Pre-LN     | -14.18   | -2.18    | 0.34     | -1.34    | -0.03    | -0.25    | 0.26     | -0.16    | -1.34    | -0.41    | -0.08    | -1.17    |
>     | Post-LN    | -1.13    | -1.17    | -1.47    | -1.38    | -1.38    | -1.30    | -1.97    | -0.63    | -1.34    | -2.23    | -3.49    | -7.49    |
>
>
> **Q2: Is this approach still valid for other tasks (like for example image classification, with a Transformer architecture on ImageNet-1k)?**
>
> - To evaluate Mix-LN on non-language tasks, we replaced Pre-LN in ViT models with Mix-LN with α=0.25 and trained the updated model on ImageNet-1K, following the ConvNeXt training configurations for 120 epochs. The results clearly demonstrate that the benefits of Mix-LN also generalize to vision tasks. Notably, the performance gains are more pronounced for larger model (ViT-Small) than smaller model (ViT-Tiny).
>
>     |  | Pre-LN | Mix-LN |
>     | :---- | :---- | :---- |
>     | ViT-Tiny | 67.30 | **67.34** |
>     | ViT-Small | 75.99 | **76.40** |

---

> ### Author Response · Authors · 2024-11-20
>
> ### **Response to Reviewer iHrq [3/3]**
>
> **Q3: Have the authors attempted to compare with other normalization techniques (like group normalization)?**
> - As requested, we conducted comparisons using LLaMA-250M to evaluate Mix-LN against recent normalization methods, including Admin [1], Sandwich-LN [2], and Group-LN [3,4]. The results indicate that Sandwich-LN and Group-LN slightly outperform Pre-LN, while Admin performs worse. However, all of these methods fall to reduce perplexity below 23, falling short of Mix-LN. This result highlights the effectiveness of Mix-LN compared to other recent innovations.
>     | Model         | Pre-LN | Admin  | Group-LN | Sandwich-LN | Mix-LN  |
>     | :------------ | :----- | :----- | :------- | :---------- | :------ |
>     | LLaMA-250M    | 23.39  | 24.82  | 23.10    | 23.26       | **22.33** |
>
> **Q4: Can the authors provide theoretical boundaries to the vanishing/exploding gradient conditions for equations (3) and (4)?**
> - Thank you for your insightful question. The theoretical analysis of gradient vanishing for equations (3) and (4) has been extensively studied in prior works. Many of the conclusions we rely on are based on the findings of Xiong et al. (https://arxiv.org/pdf/2002.04745), who demonstrated that the gradient magnitude through Layer Normalization (LN) is inversely proportional to the magnitude of its input (as stated in equation (5) of our paper). Subsequently, Liu et al. (https://arxiv.org/pdf/2004.08249) provided a more detailed theoretical analysis of gradient vanishing issues for both Pre-LN and Post-LN in Appendix A of their paper.
>
> - More recently, a great paper from Sho Takase et al. (https://arxiv.org/pdf/2312.16903) presented theoretical boundaries for the gradient norm across layers, offering further insights into this topic. Given these comprehensive studies, we kindly refer the reviewer to these excellent works for a deeper understanding of the theoretical analysis, rather than redundantly reinventing the wheel.
>
>     **Reference**
>
>     [1] Liu, L., Liu, X., Gao, J., Chen, W. and Han, J., 2020. Understanding the difficulty of training transformers. arXiv preprint arXiv:2004.08249.
>
>     [2] Ding, M., Yang, Z., Hong, W., Zheng, W., Zhou, C., Yin, D., Lin, J., Zou, X., Shao, Z., Yang, H. and Tang, J., 2021. Cogview: Mastering text-to-image generation via transformers. Advances in neural information processing systems, 34, pp.19822-19835.
>
>     [3] Wu, Y. and He, K., 2018. Group normalization. In Proceedings of the European conference on computer vision (ECCV) (pp. 3-19).
>
>     [4] Ma, X., Yang, X., Xiong, W., Chen, B., Yu, L., Zhang, H., May, J., Zettlemoyer, L., Levy, O. and Zhou, C., 2024. Megalodon: Efficient llm pretraining and inference with unlimited context length. arXiv preprint arXiv:2404.08801.

---

> > ### Comment · Reviewer_iHrq · 2024-11-25
> >
> > I thank the authors for their quite extensive rebuttal. Considering also the other reviewer's points, I see there are other concerns related to this work, but overall I am happy to remain with my borderline acceptance score.

---

> ### Author Response · Authors · 2024-11-25
>
> Dear Reviewer iHrq,
>
> We sincerely thank you for your insightful feedback and support. Your constructive comments have been instrumental in refining our work!
>
> Best regards,
>
> The Authors

---

### Comment · Area_Chair_PG2b · 2024-11-28

Dear Reviewers,

If you have not responded to author's rebuttal, please kindly do so as soon as possible. The deadline is Dec 2, but the authors can potentially further clarify questions if you respond earlier. Thanks!

Best, AC

---

### Meta-Review · Area_Chair_PG2b · 2024-12-08

**Metareview:**

Summary: Mix-LN combines pre-LN and post-LN to tackle gradient issues; it improves deeper layers' learning and leads to better overall model performance.

Strengths: motivation on balancing gradient dynamics; extensive experiments on LLMs show improvement; improved stability

Weaknesses: benefits on large models diminish; not a groundbreakingly novel method; presentation clarity issues; limited non-LLM eval

Reasons for decision: overall wide support from reviewers; simple method that lead to improvement across settings outweighs flaws.

**Additional Comments On Reviewer Discussion:**

The authors addressed scalability concerns with additional experiments on LLaMA-7B, included comparisons with other normalization layers, and improved figure clarities. These responses resolved several reviewer concerns, leading to one reviewer upgrading the rating.

---

### Decision · Program_Chairs · 2025-01-22

Accept (Poster)